# Clove Essential Oil: Chemical Profile, Biological Activities, Encapsulation Strategies, and Food Applications

**DOI:** 10.3390/antiox13040488

**Published:** 2024-04-19

**Authors:** Rafael Liñán-Atero, Fatemeh Aghababaei, Samuel Rodríguez García, Zahra Hasiri, Dimitrios Ziogkas, Andres Moreno, Milad Hadidi

**Affiliations:** 1Department of Organic Chemistry, Faculty of Chemical Sciences and Technologies, University of Castilla-La Mancha, 13071 Ciudad Real, Spain; rafael.linan@uclm.es (R.L.-A.); samuel.rodriguez@uclm.es (S.R.G.); dimitrios.ziogkas@uclm.es (D.Z.); 2Aora Health, Scientific Park of Madrid, Faraday 7, 28049 Madrid, Spain; 3College of Veterinary Medicine, Islamic Azad University of Shahrekord, Shahrekord 88137-33395, Iran; zahra.hasiri@iau.ac.ir; 4Department of Physiological Chemistry, Faculty of Chemistry, University of Vienna, 1090 Vienna, Austria

**Keywords:** clove essential oil, bioactive compounds, extraction methods, encapsulation strategies, biological activity

## Abstract

Plants have proven to be important sources for discovering new compounds that are useful in the treatment of various diseases due to their phytoconstituents. Clove (*Syzygium aromaticum* L.), an aromatic plant widely cultivated around the world, has been traditionally used for food preservation and medicinal purposes. In particular, clove essential oil (CEO) has attracted attention for containing various bioactive compounds, such as phenolics (eugenol and eugenol acetate), terpenes (β-caryophyllene and *α*-humulene), and hydrocarbons. These constituents have found applications in cosmetics, food, and medicine industries due to their bioactivity. Pharmacologically, CEO has been tested against a variety of parasites and pathogenic microorganisms, demonstrating antibacterial and antifungal properties. Additionally, many studies have also demonstrated the analgesic, antioxidant, anticancer, antiseptic, and anti-inflammatory effects of this essential oil. However, CEO could degrade for different reasons, impacting its quality and bioactivity. To address this challenge, encapsulation is viewed as a promising strategy that could prolong the shelf life of CEO, improving its physicochemical stability and application in various areas. This review examines the phytochemical composition and biological activities of CEO and its constituents, as well as extraction methods to obtain it. Moreover, encapsulation strategies for CEO and numerous applications in different food fields are also highlighted.

## 1. Introduction

Aromatic plants have a historical legacy of being employed for their preservative and medicinal attributes, as well as for enhancing the aroma and flavor of food. These characteristics are, in part, ascribed to essential oils (EOs), which are natural, intricate, and multi-component systems predominantly comprising terpenes alongside various non-terpene components [1]. EO components are biosynthesized by plants as secondary metabolites to protect themselves from pathogens and predators [2], but they also have gained attention because (a) they can be employed in preservation techniques, (b) these substances are widely acknowledged as safe, and (c) they exhibit various characteristics that hold potential for their application as bioactive compounds in diverse food products [3]. Clove, scientifically known as *Syzygium aromaticum* (L.) Merrill. & Perry, belonging to the Myrtaceae family, is a pivotal spice and ranks as the second most valuable in global trade. The clove plant, an evergreen tree that can reach a height from ten to twenty meters [4], is widely cultivated in the Maluku Islands (Indonesia), although it can also be found in other places such as Brazil, Egypt, Madagascar, and Morocco [5]. Three EOs are obtained from the clove tree: clove stem oil (flower stalks), clove leaf oil (leaves), and clove bud oil (flowers bud), the latter being the most used. EOs obtained from aromatic plants such as cloves represent a rich and varied source of natural products, including bioactive compounds, which are commonly utilized in pharmaceutical, cosmetic, health, and food sectors for their bioactive properties [6]. Due to the increasing interest of consumers in natural ingredients and their concern about synthetic and potentially hazardous additives, the global supply of EOs is currently experiencing significant growth [7]. Clove essential oil (CEO) has traditionally been used to treat wounds and burns, as well as in the treatment of dental infections and toothache [8]. In addition, several reports have documented other bioactivities of CEO, such as antiviral [9], antimicrobial [10], and antifungal and antioxidant [11], among many others. The effective role of CEO in the treatment of various pathologies is attributed to the presence of several chemical components. In this sense, it has been described that the major compound in CEO is eugenol (at least 50%), followed by β-caryophyllene (5–15%) and to a lesser amount of α-caryophyllene (also called α-humulene) and acetyl eugenol [12,13,14]. Nevertheless, each CEO may differ in its chemical composition and bioactivity due to different factors such as the agroecological conditions, the genetic factors, the variety, or the extraction processes employed to obtain it [15].

Common extraction methods for CEO and its bioactive compounds include steam distillation (SD) [16], hydrodistillation extraction (HD) [17], and soxhlet extraction (SE) [18]. Although these methods have been traditionally used, they also present some limitations such as the loss of volatile compounds by thermal degradation or the use of organic solvents, which can leave toxic residues in the extract. In order to avoid these problems, some authors have used more efficient and environmentally friendly separation techniques like supercritical fluid extraction (SFE) [19], ohmic heating-assisted hydrodistillation (OAHD), ultrasound-assisted extraction (UAE) [20], or microwave-assisted extraction (MAE) [21].

The resulting EO can be used for numerous applications in various industries. However, using or storing CEO in its conventional form poses technical challenges, as it is susceptible to degradation due to environmental stress, the presence of unstable volatile compounds, typical processing, and storage conditions. Moreover, its limited solubility in water hinders its incorporation into aqueous matrices, leading to additional limitations, since large amounts of CEO are required to exert an inhibitory effect on bacteria [22]. Fortunately, encapsulation processes could solve these problems, since the incorporation of EOs in an encapsulation matrix can prolong the shelf life of the CEO, improving its physicochemical stability, which allows it to propose new applications and increase its added value. Examples of encapsulation techniques include spray drying, ionic gelation, emulsification, and molecular entrapment, among others [23]. This review will focus on the study of the CEO and its main compounds, in addition to describing the current extraction methods available and its bioactivity. Potential applications derived from its use in different industries will be highlighted, as well as the strategies that exist for its encapsulation.

## 2. Main CEO Components and Their Bioactivities 

The EO of *S. aromaticum* has long been used in various fields such as folk medicine or the food industry due to the presence of various bioactive compounds and their properties. Several extraction methods have been used to obtain CEO, although the use of each technique may result in different chemical compositions in the final product. Table 1 shows the list of chemical components present in the CEO (expressed in percentages) obtained by three different extraction techniques: SFE, HD, and SD. All CEOs were isolated from flower buds and analyzed by GC-MS to determine their chemical composition. A total of 13 compounds were identified in the CEO, with the highest number of compounds identified in the EO obtained using SFE. Eugenol was found to be the major compound (55.28%), followed by (E)-β-caryophyllene (20.97%) and α-humulene (7.08%). Eugenol was also the main component for the rest of the CEOs (82.16–97.80%), a result that agrees with the observations of other studies. Kennouche et al. [24] isolated CEO using conventional HD, MAE, and microwave steam distillation (MSD) finding that eugenol was the major compound in all cases (up to 71.84%). Similarly, de Oliveira et al. [19] conducted an analysis of the chemical composition of CEO obtained through SFE, revealing concentrations of 2.7%, 3.1%, 24.8%, and 66.9% for eugenol acetate, α-humulene, β-caryophyllene, and eugenol, respectively (Figure 1). 

Eugenol, identified as 4-allyl-2-methoxyphenol, is an aromatic compound categorized within the phenols group, specifically a phenylpropanoid. It is characterized by its slightly yellow color, its oily consistency, and its pungent aroma [29]. Because of its excellent biological properties, numerous studies have focused on eugenol. For instance, a study conducted by Jung et al. [30] explored the effect of eugenol on Alzheimer’s disease (AD) in eight mice according to the 5× FAD mouse model. The primary findings indicated that the application of eugenol effectively reduced cognitive impairment, demonstrating notable effects in alleviating inflammation and necroptosis. These are crucial targets for achieving an improved prognosis in AD. It has also been described that eugenol can be active against various microorganisms, since it can break down the outer membrane of bacteria [31]. In this regard, Bai et al. [32] observed that eugenol effectively inhibited *Staphylococcus aureus* biofilm formation; this could be attributed to the damage inflicted on the cell wall and membrane, the hindrance of biofilm formation, and the induction of apoptosis alongside disruption in DNA synthesis. It has also been shown to possess antifeedant effects against the larvae of *Spodoptera frugiperda*, a pest of corn (*Zea mays* L.) [33]. Additionally, researchers found that eugenol may positively mediate the GABA_A_ receptor, exerting anesthetic and analgesic effects [34]. 

Taking into account the bioactive effects of eugenol, the research of some eugenol derivatives has attracted attention. Specifically, acetyl eugenol has been studied for its biological properties, including its antioxidant, anti-inflammatory, and antimicrobial activities [35]. In a study by Abdulhamid et al. [36], different fractions were isolated from the leaves of *Acacia nilotica* L. and tested for antimicrobial activity against seven bacterial clinical isolates: *Escherichia coli*, *Klebsiella pneumoniae*, *Proteus* sp., *Pseudomonas aeruginosa*, *S. aureus*, *Salmonella typhi*, and *Streptococcus pneumonia*. The results showed that fraction I, whose main component was acetyl eugenol, was effective against all the species tested, especially highlighting its antimicrobial potential against *Proteus* sp., *S. typhi*, *P. aeruginosa*, and *S. pneumonia*. Another study by Kaur et al. [11] confirmed the antimicrobial potential of acetyl eugenol by testing it against the fungus *Rhizoctonia solani*, observing 100% inhibition of mycelial growth. On the other hand, He et al. [37] evaluated the insecticidal potential of eugenol and various derivatives against *Solenopsis invicta*, concluding that both eugenol and acetyl eugenol (among others) showed strong repellent ability by significantly decreasing ant numbers in the tested scenarios. 

Other components present in CEOs, such as β-caryophyllene and α-humulene, belonging to the sesquiterpenes group, have shown bioactive effects as well. In a study conducted by Kamikubo et al. [38], it was discovered that β-caryophyllene inhibits lipid accumulation induced by palmitate in human HepG2 hepatocytes. This suggests its potential in preventing and improving non-alcoholic fatty liver disease (NAFLD) and related metabolic disorders. Moreover, Ullah et al. [39] evaluated the antioxidant and antibacterial properties of β-caryophyllene as a potential food packaging agent. Zein/polycaprolactone nanofibers loaded with β-caryophyllene/halloysite provided long-term sustained release of β-caryophyllene, showing good DPPH free radical scavenging and effectiveness against Gram-positive *Bacillus subtilis* and Gram-negative *E. coli* bacteria. Another study revealed the anti-inflammatory and anti-ulcer potential of β-caryophyllene [40]. The compound was orally administered to mice with induced colitis, showing that the disease improved by decreasing myeloperoxidase enzyme activity, the proinflammatory agents (IL-1β, IL-6, and TNF-α) concentrations, and the abundance of bacteria causing intestinal immune imbalance. On the other hand, the monocyclic sesquiterpene α-humulene showed anti-inflammatory and anti-ulcer activities as well. The administration of this compound in rats significantly reduced the lesions produced by induced acute gastritis. In addition, it also reduced histamine levels, whose excessive secretion is the main cause of gastric ulcers. Another study confirmed the anti-inflammatory potential of α-humulene since it reduced allergic airway inflammation in mice, an effect that appeared to be mediated by the reduction of inflammatory mediators [41]. Furthermore, the antioxidant effect of α-humulene was also examined. A research conducted in rats demonstrated that α-humulene effectively reduced levels of reactive oxygen species and malondialdehyde by modulating prostaglandin E2 (PGE2) and superoxide dismutase (SOD). This indicates that α-humulene displays protective effectiveness against experimentally induced gastritis through diverse mechanisms [42].

Nevertheless, the bioactivity of one EO may differ from another depending on several factors such as plant location, soil type, or climatic factors [15]. According to Table 1, although hydrodistillation was used to obtain CEOs from Morocco and Brazil, the chemical composition and amount of each compound was different for each oil, which directly affects its bioactivity. In this context, Benali et al. [43] provided an elaborate account of both the chemical composition and biological effects of EOs extracted from *Lavandula stoechas*, gathered from three distinct locations in Morocco. Their findings substantiate the theory that the synthesis of secondary metabolites in a given plant is quantitatively and qualitatively influenced by climatic, environmental, and nutritional conditions, along with external factors like humidity or temperature. Likewise, according to Pastare et al. [44], the quantitative levels of compounds isolated from a complex sample cannot be directly compared with those reported in other studies. This variability is attributed to the factors mentioned earlier, along with the extraction techniques employed and material processing.

In this regard, several methodologies have been used to obtain EOs from plant matrices. SFE has been recognized as an environmentally friendly and clean extraction method with no negative impact on nature. It is widely used to produce EOs, since it has demonstrated high selectivity toward lipophilic compounds [45]. Additionally, carbon dioxide is frequently employed as an extraction solvent owing to its physical and chemical attributes, including a low critical pressure (74 bar) and temperature (32 °C). It is non-toxic, non-flammable, readily available in high purity at a reasonably low cost, and easy to eliminate from the extract [46]. There is a plethora of studies that have explored the potential of SFE. For instance, a study by Luca et al. [47] conducted a study examining how various combinations of pressures and temperatures affected the extraction of β-caryophyllene, α-humulene, caryophyllene oxide (terpenes), and cannabidiol from hemp flowers using supercritical CO_2_ extraction (SFE-CO_2_). The findings indicated that within the pressure range of 90 to 110 bar and temperatures between 40 to 50 °C, rapid and effective extraction of the three terpenes occurred. In contrast, higher pressures (200–300 bar) were necessary to achieve elevated cannabidiol content. In the same way, Pastare et al. [44] employed SFE-CO_2_ using ethanol as co-solvent to obtain extracts from the white stripe flowers of *Matricaria chamomillo*. They also investigated how changes in pressure and temperature affected extraction yield and secondary metabolite groups (phenols, flavonoids, tannins, and sugars), concluding that both the temperature and the extraction pressure affected extraction yield, while only extraction pressure influenced the content of secondary metabolite groups. 

Besides SFE-CO_2_, there are other techniques widely used to obtain EOs such as steam distillation and hydrodistillation [48]. HD and SD have been conventionally used for the extraction of plant components. While these techniques are relatively safe and straightforward, a drawback is their time-consuming nature and the need for substantial quantities of organic solvents. In addition, the extraction efficiency is normally poor due to the degradation of thermolabile compounds [49]. For instance, Kessler et al. [50] subjected fresh and dried plant samples of *Rosmarinus officinalis* to the HD and SFE-CO_2_ extraction methods, noting that the extraction yields of EOs, both from fresh and dried sources using HD, were found to be inferior compared to those obtained through SFE-CO2. The same conclusions were reached by Gilani et al. [51], who used HD and SFE-CO_2_ techniques to extract EO from citron peel, observing that the highest extraction yield, amount of total phenolic compounds and flavonoids, and antioxidant activity were found in the EO obtained using supercritical CO_2_. Similarly, several researchers have evaluated the composition and yield of EOs obtained using SD and SFE-CO_2_. Ebrahimzadeh et al. [52] compared the EOs of *Zataria multiflora* Boiss isolated with SD and SFE-CO_2_, concluding that the SFE method offered many important advantages over SD: supercritical CO_2_ required shorter extraction times, and allowed for the possibility to control the oil composition by changing the extraction parameters, which resulted in higher selectivity compared to the SD method. These results are consistent with those of other studies in which SFE was found to give better yields compared to other conventional techniques [53,54]. Therefore, it seems that SFE is generally more suitable than other conventional techniques to produce EOs. A comparison of the three techniques mentioned above (SFE, HD, and SD) was carried out by Conde-Hernández et al. [55] who evaluated the oil quantity, antioxidant potential, and chemical composition of rosemary EOs. Higher yields of EO were observed in rosemary oil obtained by supercritical CO_2_, followed by SD and, finally, HD. Moreover, the antioxidant activity in the rosemary oil obtained using SFE-CO_2_ was also superior to that obtained by the other techniques. Another study by Wenqiang et al. [18] compared the CEO obtained using SFE with that obtained using HD, SD, and soxhlet extraction (SO). The variables evaluated for supercritical CO_2_ extraction were temperature, pressure, and particle size of *S. aromaticum*. The results indicated that the higher yield of oil extraction using SFE was mainly due to particle size, so that the smaller the particle size (the more crushed the buds are), the more oil is released, being more easily extractable by direct exposure to supercritical CO_2_. Furthermore, it was observed that an increase in temperature (from 30 to 40 °C) and pressure improved the extraction yield, due to the higher solubility of the oil components. On the other hand, the comparison of SFE against conventional techniques (HD, SD, SO) revealed that the use of SFE resulted in higher percentages of bioactive compounds present in CEO. 

Other researchers have also compared HD and SD. For instance, Řebíčková et al. [56] assessed the extraction yield, chemical composition, as well as the antimicrobial and antioxidant attributes of the EO derived from *Laurus nobilis* L. leaves, by employing HD and SD. The researchers concluded that HD is more effective in achieving higher extraction yields, whereas SD is preferable for obtaining superior quality oils with more robust biological properties. Conversely, Narayanankutty et al. [57] found lower yields when extracting curcuma EO with HD versus SD, obtaining yields of 1.25% and 1.44%, respectively. Therefore, it is not clear which of the two conventional extraction techniques yields better results because it depends not only on the extraction technique, but also on the plant matrix and the compounds that are extracted.

## 3. Recent Advances in the Extraction of CEO

Prior to the application and utilization of CEO, the crucial step is the extraction process, which consists of obtaining the desired compounds from the plant material. These CEOs are mainly extracted from the dried flower buds, although they can also be extracted from the leaves, and to a lesser extent from the stems [58]. The choice of the extraction method is very important as it will influence the composition and quality of CEOs, thus determining their suitability for different applications. Figure 2 shows the advantages and disadvantages of conventional and innovative techniques for the extraction of compounds from plant material. 

Traditionally, CEOs have been extracted using conventional techniques, including solvent extraction (SE), hydrodistillation (HD), and SD [59]. However, despite being the most widespread at the industrial level because of their low cost and simplicity, they have several disadvantages that distance them from the concepts of sustainable and green development. These drawbacks encompass longer extraction times, substantial energy consumption, large amounts of raw materials, and, in the case of SE, high consumption of solvents, related to a potential environmental impact [60]. Among the various solvents used for oil extraction, hexane extraction proved to be the best for its nutraceutical and pharmaceutical value. However, as a compound derived from petroleum, hexane has a negative impact on health. It is classified as a C.M.R. solvent (carcinogenic, mutagenic, and reprotoxic), and has also been found to be explosive. In addition, the application of high temperatures during hexane extraction could affect the quality of the extracted oil. The hazards of organic solvent residues limit the use of traditional extraction methods and reinforce the need to develop environmentally friendly and economical extraction methods to substitute conventional techniques [61]. Additionally, these methods are associated with issues such as hydrolysis and thermal degradation of some constituents, which could alter the taste and fragrance of the CEOs [62]. To overcome these limitations, new innovative techniques have been explored that represent a promising alternative by employing more efficient and sustainable approaches for the extraction of CEOs. Table 2 summarizes the conditions and extraction yield of both conventional and innovative techniques. However, a comparison of the different approaches in terms of yield is difficult, as the composition of clove plants may vary depending on cultivar and geographical origin [63].

### 3.1. Conventional Methods

In the SE process, EO is transferred from the raw plant using an organic solvent in which the oil is soluble due to their lipophilic and hydrophobic nature, being the most common solvents n-hexane, ethanol, or acetone [70]. For the procedure, plant samples are usually mixed with the chosen solvent by slightly heating for long time, and the process is followed by evaporation and filtration of the solvents [71]. A variant of this method is the Soxhlet extraction, in which clove buds are placed in a cellulose cartridge and progressively filled with the organic solvent from a distillation flask. A siphon operates by drawing the contents of the cartridge and releasing them back into the distillation flask once the liquid reaches the overflow level, facilitating the transfer of the extracted analytes. This process is conducted until the extraction reaches its final point [72]. Finally, the solvent is removed, but may still contain traces in the final product due to incomplete removal, thus causing allergies or toxicity [73]. For this reason, instead of this method, SD or HD are mainly used for industrial purposes.

SD is widely recognized as the primary choice for large-scale commercial extraction of EOs from aromatic plants. Approximately 90% of EOs are obtained through the utilization of this technique [74]. During the process, the separation of plant material from water is undertaken to safeguard the non-volatile components and prevent their depletion. Steam is used as extracting agent to vaporize the high boiling point volatile compounds present in the plant material at a temperature near 100 °C [58]. The applied hot steam is the main cause of the bursting and rupture of the cellular structure of the plant matrix. Consequently, EOs are released from the plant material until they reach a cooled condenser that causes the vapor to condense again into a liquid, formed by two immiscible phases: water and EOs [73,75], which can be separated using a Clevenger apparatus, a separation funnel, or other equipment [76].

HD is another of the most common methods and consists of immersing the plant material in boiling water, characterized by direct contact between them. The volatile compounds present in the raw material are vaporized together with the water vapors, which are separated after condensation and decantation, as in the SD process [58]. In comparing these two distillation processes with respect to yield, distillation speed, loss of oxygenated compounds, and hydrolysis susceptibility, it is possible to consider that SD has higher yields and lower susceptibility to hydrolysis, but the HD process is faster [77].

### 3.2. Innovative Methods

#### 3.2.1. Supercritical Fluid Extraction 

SFE is an alternative method for CEO extraction. Its main peculiarity is the use of supercritical fluids, which, at their critical point of pressure and temperature, share the physical characteristics of both gas and liquid [58]. The extraction process entails the passage of supercritical fluid through the clove matrix, where it dissolves and transports the desired compounds [35]. These solvents boost the extraction rate thanks to their low viscosity and high diffusivity coefficient, enabling the solvent to quickly penetrate the matrix. Carbon dioxide (CO_2_) is widely employed due to its non-harmful properties for both human health and the environment. Additionally, its relatively moderate critical temperature (31.2 °C) allows for the safe preservation of temperature-sensitive bioactive compounds [78]. However, CO_2_ has a low polarity, making it less effective in extracting highly polar phytochemicals embedded in the cell walls of vegetable samples. To address this issue, polar solvents such as ethanol, methanol, or water are used as modifiers or co-solvents. It should be noted that no solvents remain in the final products because CO_2_ reverts to its gaseous state under atmospheric conditions [79], so SFE is a green technique, more selective, with shorter extraction times and less pollution, although its initial investment is costly [80].

As an environmentally friendly method, Wenqiang et al. [18] extracted CEO under SFE and compared it with SE, SD, and HD in terms of yield, eugenol content, extraction time, color, and texture. The higher yield was obtained from SE (41.8%), accompanied by a lower eugenol content (30.8%). This observation can be explained by the extraction of undesirable compounds alongside CEO when organic solvents are employed. On the other hand, SD and HD had a yield of 10.1% (61.2% eugenol content) and 11.5% (50.3% eugenol content), respectively. Therefore, SFE was the most successful extraction method to extract better quality oil, with a yield of 19.56% (58.8% eugenol content) at a temperature of 50 °C, a pressure of 10 MPa, and in a shorter time compared to the other methods, as can be seen in Table 2. Frohlich et al. [64] extracted CEO from leaves using SFE and SE, obtaining very low yields (1.08% and 1.90%, respectively) and with very low eugenol content (29.84% and 5.67%, respectively), so that, as discussed above, although it results in higher yields, SFE has the advantage of giving purer extracts in terms of eugenol and in a shorter time. The low yields obtained in the case of CEO extracted from leaves could be explained because this part of the clove is less rich in these volatile compounds than buds.

In other studies, such as that by Roy et al. [65], optimized the extraction conditions considering the temperature, pressure, and geometry of the extractor, obtaining a maximum CEO yield of 17.9% (72.08% eugenol content) with a temperature of 44.7 °C, a pressure of 24.5 MPa, and an extractor tube of 0.15 cm in diameter. In addition, Hatami et al. [66] combined the SFE technique with cold pressing and compared it with SFE, obtaining a yield of 22.2% (57.69% eugenol content) and 21.3% (55.44% eugenol content), respectively.

#### 3.2.2. Ultrasound-Assisted Extraction (UAE)

UAE is a green extraction technique that employs ultrasonic waves to improve the extraction in terms of time and yield of bioactive compounds from plant materials. High-frequency ultrasonic waves ranging from 20 to 100 kHz can be applied either through direct contact with the sample, using an ultrasound system coupled with a probe, or indirectly through the walls of the sample container, as in the case of an ultrasonic bath [62]. These waves generate the phenomenon of cavitation, creating bubbles on or near the surface of the sample, which implode generating enough energy to rupture the cells, causing an increase in temperature and pressure, and thus the oil diffuses into the solvent [81]. The continuation of the process consists of obtaining the EO, from the evaporation and condensation of the oil, using the conventional methods previously mentioned or innovative ones, finding mainly UAE-HD, UAE-SD, or UAE-CO_2_. Therefore, UAE provides several advantages over traditional extraction methods, such as shorter extraction times, reduced solvent consumption, and higher yields [70].

Yang et al. [67] explored the ultrasound-assisted extraction in combination with supercritical fluid extraction (UAE-CO_2_) and observed a 13.5% increase in CEO extraction yield (22.04%) employing milder operating conditions, such as CO_2_ flow rate, pressure, temperature, and time, compared to SFE. The same trend was observed by Wei et al. [68] who obtained a yield of 23.19% using UAE-CO_2_, which resulted in a large increase in yield compared to traditional methods such as SE and SD, concluding that UAE-CO_2_ can be an applicable and viable method for extracting CEO at large scale.

For the UAE-HD process, Jadhav et al. [60] optimized the extraction conditions, which can be seen in Table 2, obtaining a CEO yield of 15.23% (70.29% eugenol content), reaching the conclusion that the synergy between UAE and HD enhanced processing conditions by reducing extraction time and consequently increasing the production rate.

#### 3.2.3. Microwave-Assisted Extraction 

MAE is another emerging technique characterized by the use of microwave radiation to heat the polar molecules present in the overall plant material. When these electromagnetic waves are incident on the sample, water and other polar molecules absorb the energy, converting it into heat through processes like ionic conduction, dipole rotation, or a blend of both [58]. Consequently, this leads to an increase in temperature inside cells, resulting in the rupture of the cells and the release of volatile components to the outside medium [82]. As in the case of UAE, the process ends with the recovery of the EO through the coupling of another method, including MAE-HD or MAE-SD. In contrast to conventional methods, MAE offers several advantages, including superior heating efficiency, faster energy transfer, reduced time and energy usage, as well as higher extraction yields [62,83].

Kapadiya et al. [21] optimized the optimal conditions for MAE with three responses: CEO yield, eugenol yield and bacterial inhibition, achieving 13.11% and 11.93%, respectively, for the first two responses, with the conditions found in Table 2. This optimization allowed them to conclude that higher power and lower frequency can result in better extraction yields, but excessive heating can lead to degradation of target compounds. In addition, longer extraction times give longer yields, but there is a risk of compound degradation due to high exposure to these microwave radiations. Golmakani et al. [69] carried out CEO extraction using MAE-HD and MAE-SD, obtaining yields of 13.94% and 12.71%, respectively.

Additionally, the findings from Gonzalez-Rivera et al. [63] regarding the extraction of CEO using MAE were promising, achieving a yield of 16%. This method was described as environmentally friendly, competitive, and economically attractive. It was highlighted as a readily applicable approach for various substrates, including aromatic herbs, spices, and seeds, enabling faster and more efficient EO extraction compared to conventional techniques.

#### 3.2.4. Ohmic Heating-Assisted Hydrodistillation

The use of ohmic heating for EO extraction is a novel concept that has piqued the interest of researchers worldwide, being used to extract it from several plants such as clove, cinnamon, or bay leaf [84] and lavender [85]. OAHD is an environmentally friendly technology that can be scaled up to the industrial level to obtain these EOs [86]. The principle is based on combining the HD technique with ohmic heating, which is the mechanism whereby the electrical resistance of the plant material itself generates heat by passing an electric current through it, thus causing the rupture of its plant cells and therefore the release of the compounds of interest [58]. In this sense, OAHD has potential advantages, such as saving extraction time and less energy consumption by producing heat directly inside the electroconductive plant material [76]. In addition, compared to HD or MAE, the carbon footprint is lower by consuming less power [87], making it one of the most energy-efficient, clean, and environmentally friendly methods of EO extraction. However, a high capital investment is required, which hinders its industrial implementation [85].

Tunç and Koca [20] optimized the extraction of CEO with OAHD using three variables: applied voltage, time, and clove mass. CEO yield of OAHD (13.18%) was higher than that of the HD method (8.23%), and the eugenol yield was also higher in OAHD than in HD (6.95% and 5.16%, respectively). 

It should be noted that other studies have combined several emerging ones, such as Zhang et al. [88], who combined ultrasound with ohmic heating to extract EO from citronella, observing a remarkable reduction of the energy used and a higher yield rate, so it could also be applied to cloves, although there are no studies about it yet.

## 4. Biological Activities of the CEO

CEO exhibits a spectrum of biological activities (Figure 3). Its notable antioxidant properties, stemming from a rich phenolic content, contribute to cellular protection against oxidative stress. Additionally, the oil showcases significant antimicrobial activity, predominantly due to the presence of eugenol, which inhibits the growth of various bacteria and fungi. Studies also suggest potential anti-inflammatory effects, with bioactive compounds like eugenol implicated in mitigating inflammatory responses. While research on its anti-cancer properties is ongoing, preliminary studies indicate a potential inhibitory effect on cancer cell growth. Moreover, there are indications of anti-hypertensive effects, but further research is required for conclusive evidence. These diverse biological activities highlight the multifaceted therapeutic potential of CEO, making it a subject of interest for various health applications.

### 4.1. Antioxidant Activity

A comprehensive comprehension of oxidative stress and its involvement in the pathogenic processes of various disorders is crucial [89]. This understanding paves the way for the identification of effective natural antioxidants with elevated safety and bioavailability, capable of neutralizing reactive oxygen species (ROS). This, in turn, helps maintain genomic stability while concurrently preventing disorders associated with free radicals and ageing [90]. Research has provided evidence that a variety of natural antioxidants are both safe and efficacious in reducing oxidative damage, resulting in the delay of disease onset or an enhancement of health in numerous medical conditions [91]. The worldwide manufacturing and application of EOs as natural components in food and flavorings are expanding, with over 3000 unique EOs documented from diverse plant species, components, and geographical areas [92]. The chemical composition of EOs frequently features two or three major compounds, constituting a significant portion of the product, up to 70%. These major compounds play a pivotal role in shaping the properties of EOs, notably influencing their antimicrobial, antioxidant, and biological attributes [93]. Nevertheless, the antioxidant capabilities of certain EOs may not be entirely accounted for in the presence of specific predominant constituents. Instead, these properties can be ascribed to the synergistic interactions between these major components and the minor ones [94]. In general, antioxidant capacity involves a variety of mechanisms. These include free radical scavenging, delaying free radical production and the formation of secondary toxic species, changing free radicals into less toxic compounds, interrupting the chain propagation reaction, and boosting the endogenous antioxidant defense system (Figure 4). All of these mechanisms are related to the anti-radical properties of some EO constituents and their ability to scavenge radicals, which decreases oxidative stress and in turn prevents lipid, protein, and DNA damage [95].

#### 4.1.1. Free Radical Scavenging

It has been reported that the antioxidant activity exhibited by CEO could be related to the presence of eugenol and other extracted phenolic compounds [16], which, like its other components, show activity in the 2,2-diphenyl-1-picryl-hydrazyl (DPPH) assay [96]. Thus, Kang and Song [97] investigated whether the stability and properties of pork belly could be improved when Job’s tears starch (JTS) films enriched with CEO were applied to pork belly. The assays they conducted to measure the antioxidant potential were focused on studying the potential of the CEO-enriched films to scavenge free radicals (studied by ABTS and DPPH methods), as well as their ability to inhibit lipid oxidation. JTS films enriched with different concentrations of CEO (0.25%, 0.50%, 0.75%) showed a radical scavenging effect as the concentration of CEO in the film increased, although there was no significant difference (*p* > 0.05) between 0.5% and 0.75% CEO in the case of the ABTS assay. For this reason, the authors chose the JTS film with 0.5% CEO for pork belly packaging. Subsequently, to determine the effect of CEO-JTS films on the inhibition of lipid oxidation in pork belly during storage, the peroxide value (POV) and thiobarbituric acid reactive substances (TBARS) value were measured. POV can be used as an indicator of primary lipid oxidation, while TBARS is one of the methods to analyze lipid oxidation in meat, especially secondary oxidation [98]. The results showed that the presence of CEO in the film at the aforementioned concentration caused a decrease in POV and TBARS values with respect to the controls, suggesting that CEO-JTS films have potential as an antioxidant food packaging material to improve the freshness and quality of pork belly [97].

In the same way, Xu et al. [99] evaluated the antioxidant effectiveness of CEO in the production of dry-cured duck at 60 °C for 56 days. At the outset, the POV and acid value of the duck grease containing clove extract were 0.1 g/100 g and 0.2 mg/g. On the 56th day, in the presence of CEO, these values increased to 0.52 g/100 g and 1.59 mg/g, respectively. In contrast, the POV and acid number of the control (in the presence of tert-Butylhydroquinone, TBHQ) increased to 0.90 g/100 g and 2.23 mg/g at the end of the 56th day. These results indicate that clove possesses a distinctive antioxidant capability, nearly on par with that of TBHQ. Other researchers have also evaluated the ability of CEO to scavenge free radicals. Radünz et al. [16] conducted a study to compare the antioxidant activity of unencapsulated and encapsulated CEO. They reported that unencapsulated CEO exhibited a 94.86% DPPH scavenging capacity at a concentration of 484.7 μg/mL, while the encapsulated CEO showed lower antioxidant activity for evaluated radicals, possibly due to the strong interaction between phenolic compounds and the wall material. On the other hand, at a lower concentration of 12.25 µg/mL, CEO displayed reduced inhibitions of 28.83% and 22.13% for hydroxyl and nitric oxide radicals, respectively. The authors noted that the significant DPPH scavenging activity noticed in CEO can be attributed to a synergistic interaction among its phenolic compounds, even when present in low concentrations. Conversely, the reduced inhibitory results recorded for hydroxyl and nitric oxide radicals might be ascribed to the limited interaction of the phenolic compounds with these radicals. In any case, the CEO investigated exhibited a more potent DPPH-scavenging activity in comparison to the findings documented in the existing literature. As an example, Sebaaly et al. [100] documented a scavenging rate of 92.82% at a concentration of 10,000 μg/mL. Kiki [101] also analyzed the antioxidant capacity of CEO as a function of the radical scavenging activity. In the time frame of 30 to 120 min, the inhibition percentages of CEO were assessed across five concentrations ranging from 50 to 800 µg/mL. The findings demonstrate that the CEO is highly effective at scavenging DPPH radicals. Noticeably, its efficacy exhibited a gradual rise with both higher concentration and extended duration. The highest antioxidant capacity was observed at 800 µg/mL, reaching a remarkable 98.6% in free-radical scavenging. Overall, at various concentrations, all levels demonstrated a significant antioxidant effect.

#### 4.1.2. Inhibition of Free Radical Production and Disruption of the Chain Reaction Propagation

Protective mechanisms exist to remove and detoxify ROS and block their production. A study by Zhang et al. [90] examined the effects of CEO on antioxidant activity and life expectancy extension in the nematode *Caenorhabditis elegans* (*C. elegans*), which has been identified as a favorable animal model to investigate the impacts of neural molecules and herbs in vivo. The researchers exposed C. elegans to different concentrations of CEO (0.5, 1, and 3 mg/mL) and observed that all CEO treatments induced DAF-16/FoxO nuclear translocation from the cytoplasm to the nucleus. DAF-16/FoxO is a transcription factor in which nuclear translocation is associated with the inhibition of free radical production and disruption of chain reaction propagation through the regulation of antioxidant genes and modulation of cellular responses to stress, thereby promoting cellular health and resistance to ageing [102]. In another study, the antioxidant (or anti-inflammatory) activity of pure CEO and its pharmaceutical delivery systems against croton oil-induced skin inflammation in mice was assessed [103]. The anti-inflammatory activity of the investigated formulae was assessed by employing croton oil in mice. The topical application of croton oil triggers significant inflammatory responses (oedema, vascular permeability increment…) besides several inflammatory mediators that involve overexpression of COX-2, an enzyme robustly implicated in the pathogenesis of inflammatory diseases. The authors developed a CEO-enriched nanoemulgel (formulation combining features of nanoemulsions and gels) using Taguchi’s model to optimize the formulation. In addition, scaffold-based nanofibers were also employed, which can contribute to improving the structure and stability of the formulation, as well as improve the efficacy of the delivery of active substances through the skin. According to the results, the potentiated anti-inflammatory activity of such medicated formulae (CEO-NE-based NEG and CEO-NE-based NFs) was clearly established. Both inflammation and oxidative stress are interconnected in many biological responses. Inflammation can generate ROS and other free radicals, contributing to oxidative stress. COX-2, being involved in mediating the inflammatory response, may be linked to ROS generation. Therefore, the inhibition of COX-2 could reduce the production of inflammatory proteinoids and, consequently, the generation of ROS [104].

#### 4.1.3. Free Radical Conversion into Less Toxic Compounds

Certain cellular indicators are crucial for assessing the efficacy of cellular protection against oxidative stress, such as the activity of antioxidant enzymes or intracellular glutathione levels, among others [105]. Among the enzymes that prevent or mitigate oxidative stress are superoxide dismutase (SOD), catalase (CAT), and glutathione peroxidase (GPx) (Figure 5) [106].

The source of ROS in erythrocytes is the oxygen carrier protein hemoglobin (Hgb), which undergoes autoxidation to produce superoxide anion $\left(O_{2}^{•-}\right)$. Superoxide anion is converted to O_2_ and H_2_O_2_ by SOD, a metal-containing enzyme. SOD is a family of enzymes comprising Cu-SOD, Zn-SOD, Mn-SOD, and extracellular SOD, whose function is protection against ROS, in particular $O_{2}^{•-}$ [106]. Several authors have investigated the possible relationship of these enzymes with the addition of compounds with antioxidant activity. For example, Li et al. [107] investigated the effects of CEO on the activity of some enzymes of *Varroa destructor*, a parasitic mite of honey bees. In their study, a group of *V. destructor* mites (divided into three groups) was fed with five worker bee larvae in each group. To the first group (group A, negative control), no treatment was applied, while to groups B and C, 0.1 µL and 1.0 µL of CEO were added, respectively. The results showed that SOD activity significantly increased after CEO treatments. The addition of 0.1 µL of CEO produced the greatest increment in SOD activity, reaching 40.1 ± 5.23 U/mg protein, while the untreated group presented a SOD activity of 28.52 ± 0.65 U/mg protein. This result suggests that the antioxidant reaction of *V. destructor* is related to the presence of CEO [107]. On the other hand, it has also been described that excessive accumulation of H_2_O_2_ is related to disorders such as phagocytosis, ageing, inflammation, tissue repair, and intracellular messenger pathways [106]. There are two mitigation pathways to convert H_2_O_2_ into other harmless substances. The first one is related to CAT, an enzyme that converts H_2_O_2_ to generate H_2_O and O_2_. The other one is the glutathione peroxidase (GPx), where GPx utilizes glutathione (GSH) as an electron donor to convert H_2_O_2_ to H_2_O [108]. The activity of these enzymes and their relationship to the presence of the CEO was investigated by Mohammadi [109]. In this study, the author observed the effect that different levels of CEO exerted on SOD, GPx, and CAT activities (among biomarkers of oxidative stress) in comparison with vit E. The experimental part consisted of the random division of 288 broilers into six groups (G) that received different treatments and diets: (G1) normal control (NC), in which the broilers in this group were kept in the recommended conditions and received the basal diet; (G2) heat stress control (HSC), in which the animals in this group were heat stressed and received the same diet as group 1; (G3) standard treatment (ST), in which animals in this group were also heat stressed and received the basal diet supplemented with 100 ppm vit E as a standard commercial antioxidant; and (G4–6) CEO treatment (HT), in which animals in these groups were heat challenged and received the basal diet supplemented with 250, 350, and 450 ppm CEO. Heat stress may significantly decrease animal performance and induce severe oxidative/nitrosative stress. The comparison of the results obtained for groups 1 and 2 revealed that heat stress significantly increased the activities of all three enzymes. This result was also observed in a study by Altan et al. [110], who explain that this increment could be the result of a compensatory mechanism. Since heat stress produces oxidative stress, it has been suggested that enzyme activities are enhanced to cope with oxidative stress and alleviate the harmful effects of oxidative molecules [109,110]. On the other hand, the comparison between group 2 (no dietary treatment) and groups 3–6 showed that both Vit E and CEO had antioxidant properties. Both treatments reduced oxidative stress and, consequently, the activity of the enzymes studied. This finding suggests that CEO has a potential modulatory effect on the antioxidant response of broilers, highlighting its ability to influence the activity of key enzymes that counteract oxidative stress [109].

#### 4.1.4. Delaying the Formation of Secondary Toxic Species

Slowing down the development of secondary toxic species plays a critical role in minimizing cellular harm and fostering general cellular well-being [105]. A study carried out by De Oliveira et al. [111] determined the effect that CEO and its three main components (eugenol, β-caryophyllene, and acetyl eugenol) could have as antioxidant supplements on bovine in vitro production (IVP). Bovine embryos IVP has gained popularity over the past several decades, generating more than 1.5 million bovine embryos in 2021. However, it has been described that cellular stress and/or imbalances in biological systems can harm DNA, proteins, and lipid bilayers, thus compromising oocyte quality and maturation potential and delaying embryonic development [108]. Regarding this description, the authors hypothesized that the optimization of culture systems by supplementation with antioxidants could improve bovine in vitro embryo production. The tests were performed in IVM medium evaluating nuclear and cytoplasmic maturation (experiment 1), bioenergetic/oxidative status (experiment 2), and developmental competence of bovine embryos (experiment 3). Regarding the first experiment, it has been described that the expansion of cumulus cell layers is an indicative factor associated with good nuclear and cytoplasmic maturation because the elimination of these cells weakens the ability of oocytes to synthesize GSH [108]. All antioxidant-supplemented groups experienced a greater degree of meiosis resumption and progression to metaphase II, although the researchers pointed out that the positive effects depended on the type and concentration of antioxidants used. In this sense, both CEO and eugenol promoted higher rates of cumulus cell layer expansion compared to other groups. In terms of oxidative status parameters, mitochondria are organelles found in the cytoplasm and are often used as markers to assess oocyte quality. Low values of mitochondrial membrane potential (ΔΨm levels) are associated with efficient ATP production, delay in excessive ROS accumulation and regulation of cell cycle and apoptosis [112]. The levels of ΔΨm were lower when antioxidant treatments were applied, a result similar to Santos et al. [113], who used 20 μg/mL CEO during bovine IVM and also observed a decrease in these levels. The third and last experiment was aimed at improving post-pregnancy outcomes. The results obtained with the groups treated with CEO, eugenol, and cysteine, a synthetic antioxidant used in most IVM studies, are remarkable due to the efficacy of these antioxidants in maintaining embryo quality. Embryonic kinetics revealed that D3 cleavage was higher in the aforementioned groups, demonstrating that the addition of antioxidants was responsible for promoting greater embryonic development from the early stages [111]. Therefore, CEO and its bioactive compound eugenol are potential options for application as an antioxidant to generate greater reproducibility and efficiency for bovine IVP. Furthermore, the use of CEO is also an interesting alternative for the reduction or delay of damage caused by oxidative stress in bovine oocytes.

#### 4.1.5. Immune System Improvement

Certain plants possess immunomodulatory properties exerting effects on various parts of the immune system on both cellular and molecular levels [114]. In the case of CEO, some studies have evaluated this issue. Elbaz et al. [115] address the impact of probiotics and CEO on growth, immune-antioxidant status, ileum morphometry, and microbiome in broilers under heat stress conditions. An amount of 300 broiler chickens were randomly distributed into four groups and fed with different diets: control group (G1) given a basal diet with no additives, (G2) given a basal diet with probiotics (2 g/kg diet), (G3) given a basal diet with clove essential oil (300 mg/kg diet), and (G4) given a basal diet with probiotics and clove essential oil. The findings of the study indicated that the mixture of probiotics and CEO significantly improved the performance of broilers subjected to heat stress. The adverse effects of heat stress on growth performance, nutrient digestibility, immune-antioxidant status, and ileal morphometry were significantly lower in broilers fed with these dietary supplements. This improvement may be due to the high potential of the combination of probiotics with CEO, which enhances a stimulatory effect on digestive enzyme activity, as well as antimicrobial, antioxidant, and immune activities. The phenolics contained in CEO may alleviate the effect of heat stress, improving intestinal health and nutrient digestion and absorption. In addition, other studies have indicated a connection between the addition of essential oils to broiler diets and the enhancement of the cell-mediated humoral immune response, which increases antibody production [116]. To conclude, evidence indicates that many physiological and pathological conditions such as aging, inflammation, and cell death are developed through the action of ROS. Although much work remains to be done, antioxidant research provides an important framework for elucidating radical scavenging activity, understanding oxidant/antioxidant interactions, and identifying signaling networks in radical metabolism.

### 4.2. Antimicrobial Activity

Pathogenic bacteria responsible for foodborne illnesses, including *E. coli*, *S. aureus*, *Salmonella Typhimurium*, and *Listeria monocytogenes*, are prevalent in our daily lives. They can be readily transmitted to humans through contamination in a wide range of foods, such as meat, dairy products, vegetables, and homemade items [117]. Over the past few years, a multitude of promising and eco-friendly tactics, including the utilization of enzymes, phages, and particularly natural compounds derived from plants, have been employed to address the issue of foodborne pathogen contamination within the food industry [118]. Spices and their EOs serve as natural flavorings, with many of them exhibiting antibacterial characteristics [119]. CEO is renowned for its antimicrobial properties against a variety of pathogenic bacteria. Its potent antibacterial attribute is primarily associated with its substantial eugenol content. It serves as a bactericidal agent, effectively combating key foodborne pathogens such as *S. aureus*, *E. coli*, *L. monocytogenes*, and *S. Typhimurium* [16,120].

Bai et al. [32] evaluated the antibacterial efficacy and mode of action of CEO against foodborne pathogens. The findings demonstrated a substantial impact of CEO on both Gram-positive (*S. aureus*) and Gram-negative (*E. coli*) bacteria. This outcome can be ascribed to the CEO’s lipophilic characteristics, enabling it to interact with lipids within the bacterial cell membrane, consequently enhancing membrane permeability [16]. Significantly, the minimum inhibitory/bactericidal concentration values for CEO in relation to *E. coli* stood at 0.64 and 1.28 mg/mL, respectively, slightly exceeding those observed against *S. aureus*, which were 0.52 and 1.04 mg/mL, respectively. In the case of Gram-positive bacteria, their cell walls consist of a single peptidoglycan layer, rendering them less effective in preventing the infiltration of antibacterial substances [121,122]. Nonetheless, Gram-negative bacteria exhibit a highly intricate cell wall structure, which encompasses outer membrane proteins, peptidoglycan layers, and outer membrane lipopolysaccharides. This complexity poses a considerable barrier to the penetration of harmful foreign molecules into the bacterial cells [123]. These results were in line with the other research, which found similar antibacterial activities in *Dodartia orientalis* L. and *Campomanesia aurea* O. Berg EOs [121,124]. In another interesting study, Sharma et al. [125] investigated the antimicrobial properties of CEO at various concentrations (1 wt%, 5 wt%, and 10 wt%) during 24 h in a PLA/PBAT (poly(lactide)/poly(butylene adipate-co-terephthalate) blend film designed for food packaging purposes. Results demonstrated that the film containing 1 wt% clove oil displayed antibacterial effects against both *E. coli* and *S. aureus* within the initial 4 h. Conversely, the film with 5 wt% clove oil exhibited antibacterial activity against *E. coli* for 12 h, after which its effectiveness waned. Notably, the 10 wt% clove oil composite film (PLA/PBAT-Clove10%) caused a substantial reduction in *E. coli* growth, decreasing from an initial count of 6.5 log CFU/mL to 4.4 log CFU/mL. Notably, the authors reported that the 10 wt% clove oil composite film exhibited exceptional antibacterial efficacy against *S. aureus*. When the 5 wt% clove oil composite film was used, the growth of *S. aureus* decreased from an initial count of 6.5 log CFU/mL to 4.5 log CFU/mL. In contrast, the 10 wt% clove oil composite film completely eradicated *S. aureus*, reducing its growth from 6.5 log CFU/mL to 0 log CFU/mL. Moreover, Mulla et al. [126] conducted research that revealed the potent antibacterial effectiveness of CEO against both *S. typhimurium* and *L. monocytogenes*.

Apart from the bacterial point of view, the presence of fungi in food, feed, and various agricultural products results in significant degradation and a range of food safety concerns. The substantial spoilage of food due to fungal contamination accounts for roughly 30% of the annual global food losses, and leads to considerable economic consequences [127]. Numerous authors have showcased the ability of CEO to combat isolated fungi from artistic materials through their antifungal efficacy [128,129,130]. CEO, along with its primary component, eugenol, which exhibits efficacy against fungal pathogens like *Botrytis cinerea*, *Penicillium expansum*, as well as various *Aspergillus* species such as *Aspergillus niger* and *Aspergillus flavus*, are being regarded as natural fungicidal solutions [131]. 

Kaur et al. [11] assessed the capabilities of CEO, its primary constituent, and its derivatives in their study against three pathogenic fungi *(Fusarium moniliforme* and *Helminthosporium oryzae*) using poison food techniques. CEO demonstrated 84.4% inhibition with ED50 values of 9.8, 11, and 28 μg/mL for *H. oryzae*. Moreover, against *F. moniliforme*, it exhibited 84.76% inhibition with ED50 values of 10.6, 11.72, and 25.56 μg/mL. In addition, another study was conducted to assess the antifungal properties of EOs and functional extracts derived from clove and pepper (*Piper nigrum* L.) for mitigating the severity of wilt caused by *Fusarium oxysporum* and *A. niger* in tomato (at three different concentrations: 350, 400, and 450 ppm) through a comprehensive set of in vitro and in vivo investigations by Muñoz-Castellanos et al. [132]. CEO decreased the growth of *A. niger* by 50% to 70%, and reduced *F. oxysporum* growth to 40%. The most effective combination for safeguarding tomato fruit against both phytopathogenic fungi in vivo was the mixture of functional extracts (FEs) from clove and pepper, combined with ethyl decanoate (FEs-C10). In agreement with the antifungal activity of CEOs, Ju et al. [133] documented that cinnamon and CEOs displayed potent antifungal properties against two mold species (*Penicillium* spp. and *Aspergillus* spp.) found in baked food, with a diameter of the zone of inhibition notably exceeding 15 mm. The variations in the fungicidal efficacy of plant EOs have been extensively established as being associated with their constituent active components, which encompass phenols, aldehydes, and ketones [117,134]. 

### 4.3. Anti-Inflammatory Activity

Inflammation, serving as a natural response to external harm or adverse environmental conditions, is a pivotal factor in the development of various chronic diseases like rheumatoid arthritis, diabetes, and cancer [135]. When inflammation becomes intense at the tissue level, it can lead to nerve damage, giving rise to pain signals that travel via neurons to the brain [136]. In recent times, knowledge about pain and its underlying mechanisms, particularly neurophysiological and neuropathic pain, has made substantial advancements [137]. For instance, there is growing evidence indicating that inflammation and the release of inflammatory mediators from injured tissues can be a source of pain [138]. Numerous drugs designed as anti-inflammatory or pain relievers, including both steroidal and non-steroidal options, are frequently employed in the management of inflammatory diseases and their associated discomfort, albeit with restricted efficacy and associated side effects [139]. Consequently, efforts to create environmentally friendly and natural drugs as a valuable reservoir of innovative therapeutics have garnered increased focus in recent times [140]. CEO is widely recognized for its role as an analgesic and anti-inflammatory agent, and historically has found traditional use in aromatherapy for alleviating conditions such as headaches, joint pain, toothaches, and as an oral antiseptic [141]. Its predominant bioactive component, eugenol, along with derivatives such as α-caryophyllene, acetyl eugenol, and α-humulene, possesses robust anti-inflammatory and antipyretic attributes. Hence, numerous pharmaceutical, medicinal, food, agricultural, and cosmetic sectors are actively investigating the utilization of clove oil as a pivotal raw material for their final products [142]. 

Kim et al. [14] investigated the anti-inflammatory activity of eugenol clove essential oil (ECEO) against resistant *Helicobacter pylori*. The obtained results for inflammation were derived from four distinct treatments, including the use of ECEO, with sodium diclofenac serving as a positive control. The purified ECEO demonstrated inhibition rates for human erythrocyte hemolysis at 53.04%, 58.74%, 61.07%, and 63.64% when tested at concentrations of 4, 8, 16, and 32 μg/L. In comparison, sodium diclofenac exhibited inhibition percentages of 63.72%, 67.49%, 69.18%, and 71.43% at the corresponding concentrations. The capacity of medications to stabilize erythrocyte membranes can extend to stabilizing lysosomal membranes, resulting in anti-inflammatory effects through alterations in the activity and release of cellular mediators, owing to the similarities between these membrane structures [143]. Moreover, Esmaeili et al. [140] investigated the anti-inflammatory impact through the paw edema test, the anti-nociceptive effect assessed via the hot plate, and the formalin test of nanoemulsion-based gels (NG) that incorporate clove and cinnamon EOs in an in vivo model. The suppression of carrageenan-induced paw edema in rats serves as a model for evaluating the effectiveness of anti-inflammatory medications [144]. Cinnamon-NG exhibited a greater suppression of rat paw edema induced by carrageenan compared to the other groups, which included clove-NG, control, and blank gel groups. The cinnamon-NG group displayed its highest inhibitory effect four hours after treatment. In the case of the clove-NG group, the inhibitory effect was greater than that of the distilled water and blank gel groups during the first and second hours of the experiment, although these differences were not statistically significant [140]. Furthermore, CEOs have demonstrated substantial potential for facilitating wound healing. In this regard, Banerjee et al. [141] explored the possible use of CEO in promoting wound healing and its anti-inflammatory properties. Its anti-inflammatory potential was examined in female Wistar rats. In this study, the topical application of the emulsion resulted in a 40–60% reduction in paw swelling induced by the carrageenan model, observed over a duration of 30–180 min when compared to untreated animals. Likewise, the emulsion displayed noteworthy wound healing capabilities, evident in both incision (wound breaking strength of 338.91 ± 5.02 g) and excision (95% wound contraction by the 16th day) models in these animals. The re-epithelization process was completed in approximately 10.67 ± 1.67 days, and the outcomes were on par with those achieved using diclofenac gel and neomycin cream, which served as positive controls [141]. The process of healing and wound closure is believed to be driven by the predominance of inflammatory markers, which primarily stimulate fibroblast cells and keratinocytes. This stimulation accelerates the development of the extracellular matrix and collagen, facilitating the formation of the skin tissue’s stroma [145].

### 4.4. Cytotoxic Activity

The human body is composed of countless cells in various developmental stages, typically dividing and proliferating in accordance with the body’s metabolic requirements [146]. The human body has meticulously regulated control mechanisms that involve signals prompting cells to cease their growth and initiate programmed cell death. Nevertheless, if these mechanisms malfunction, cells can exhibit irregular traits, such as uncontrolled growth, potentially leading to the formation of tumors [147]. Cancer is recognized as a leading global cause of mortality, marked by the unregulated proliferation of normal human cells [148]. Conventional approaches typically employed for cancer treatment include chemotherapy, radiotherapy, and surgical interventions [149]. Conventional therapies affect both cancerous and healthy cells indiscriminately. Therefore, there is a need to explore novel anticancer drugs derived from medicinal plants as potential alternatives to these traditional approaches. This can help mitigate the development of cancer cell resistance and reduce the toxicity associated with single-drug treatments [150]. Numerous researchers are directing their attention toward complementary and alternative medicines that not only mitigate adverse side effects, but also demonstrate consistent efficacy. A prevalent source of these treatments is phytochemicals derived from plants.

Phytochemicals, organic compounds originating from plants, have been associated with the lowered risk of major chronic diseases and cancer. These compounds have diverse functions, including the control of oncogene and tumor-suppressor expression in cancer cells, as well as their involvement in the induction of cell cycle arrest and apoptosis [151]. Therefore, phytochemicals could potentially serve as the primary component of future anticancer therapies. A wealth of research has indicated that bioactive compounds derived from plant sources, including those found in lesser-known species like *Syzygium*, may offer promising options as complementary and alternative treatments for cancer. Species of *Syzygium* renowned for their anticancer attributes encompass *S. aromaticum*, *S. aqueum*, *S. samarangense*, *S. cumini*, *S. jambos*, *S. campanulatum Korth*, *S. alternifolium*, *S. benthamianum*, *S. caryophyllatum*, *S. fruticosum*, *S. lineatum*, and *S. malaccense* [152].

Taha et al. [148] assessed the anticancer efficacy of extracts from *S. aromaticum* against HCT human colon carcinoma. The anti-cancer potential of clove extracts, including acetonic, dichloromethane, ethanolic, and petroleum ether extracts, was evaluated using the MTT assay. Among the clove extracts, the ethanolic extract demonstrated the highest anticancer efficacy against the HCT cell line, with an IC50 value of 2.53 µg/mL. Conversely, the dichloromethane extract of clove exhibited the lowest effectiveness against HCT cells, with a relative IC50 of 6.71 µg/mL. Additionally, the petroleum ether and acetonic extracts of clove displayed intermediate cytotoxicity against HCT cells, with IC50 values of 6.48 µg/mL and 2.91 µg/mL, respectively. According to the results, the authors concluded that the pronounced anticancer potential of clove extracts against the HCT cancer cell line reinforces the potential utility of these extracts in the development of natural anticarcinogenic agents. Abadi et al. [153] conducted a study aimed at enhancing the therapeutic properties of *S. aromaticum* L. This was achieved by transforming the EO from *S. aromaticum* L. buds into a nano-emulsion drug delivery system referred to as SABE-NE. The study focused on examining the anti-tumor and apoptotic effects of SABE-NE on human HT-29 colon cancer cells. The generated SABE-NE, with a size of 131.2 nm, triggered an apoptotic response and cell death, as evidenced by the up-regulation of Cas3 and increased SubG1 peaks. This led to a substantial reduction in the viability of HT-29 cells, while HFF cells exhibited limited cytotoxic effects [153]. In the regard of anti-cancer activity of CEO, the study involved the examination of in vitro cell migration and wound closure inhibition, as well as cellular apoptosis, using four chloroform extract fractions of clove on human A549 and H1299 cancer cell lines [154]. Flow cytometry analysis was performed on both the A549 and H1299 cancer cell lines. The findings revealed that the chloroform extract fractions of clove effectively inhibited cell migration and wound closure in both the A549 and H1299 cell lines, and induced apoptosis in the H1299 cell line. This was demonstrated by the significant reduction in wound closure percentages in cells treated with these fractions, in contrast to the control group which exhibited a 70% closure rate. The morphological characteristics of the cell nuclei treated with these fractions displayed chromatin compression, nuclear shrinkage, and the formation of apoptotic bodies. These observations strongly indicated a mode of cell death consistent with apoptosis. Hence, the findings provide confirmation that the chloroform extract derived from clove buds holds potential for lung cancer treatment.

### 4.5. Anti-Hypertensive Activity

Historically, aromatherapy and CEO have been employed to address conditions such as headaches, joint pain, toothaches, and oral hygiene concerns. Both CEO and eugenol are recognized as safe, cost-effective, and efficient analgesics. The analgesic properties of eugenol in various pain models have been extensively documented [62]. Khalilzadeh et al. [155] noted that the analgesic impact of CEO is attributed to the opioidergic and cholinergic systems. Particularly, the analgesic effect of CEO in cases of acute corneal pain seems to be contingent on cholinergic activity. The analgesic and local anesthetic properties of eugenol can be influenced by its ability to inhibit voltage-gated channels (specifically, Na^+^ and Ca^2+^ channels) and activate vanilloid transient receptor potential-1 (TRPV). Notably, the analgesic effects of CEO and eugenol closely resemble those of lidocaine. In research by Correia et al. [156], the analgesic effectiveness of CEO in fish was established. By administering concentrations ranging from 40 to 80 µL/L during invasive procedures or those with the potential to induce pain, a noticeable analgesic response in the animals was observed. This helped in mitigating the impact of noxious stimuli. CEO holds promise for application in painful procedures with the aim of reducing the impact of noxious stimuli, primarily for ethical reasons, and to safeguard the well-being of animals by preventing stress and its adverse repercussions [157]. In another study, Taher et al. [158] assessed the antinociceptive capabilities of CEO in mice. In the antinociceptive test, mice subjected to clove oil treatment displayed a notable reduction in acetic acid-induced writhing movements, reaching a maximum reduction of 87.7% (*p* < 0.01), while aspirin injection (100 mg/kg, intraperitoneal, i.p.) resulted in a 77.7% decrease *(p* < 0.01). Likewise, in the hot plate test, the administration of clove oil led to a substantial 82.3% increase in the reaction latency to pain after 60 min (*p* < 0.05), in comparison to the 91.7% increase (*p* < 0.01) observed with morphine.

Nonsteroidal anti-inflammatory drugs represent the most-employed medications for managing nociceptive pain associated with inflammation. Their primary mode of action involves the inhibition of cyclooxygenase (COX), which leads to a reduction in prostaglandins responsible for nociceptive pain. Eugenol’s antinociceptive and anti-inflammatory properties are linked to its ability to inhibit COX-2 and the TRPV by suppressing high-voltage Ca^2+^ currents in primary afferent neurons [159]. The antinociceptive effect is associated with opioid, cholinergic, and α2-adrenergic receptors, while serotoninergic receptors are not involved. Eugenol’s antinociceptive impact is likely connected to the modulation of gamma-aminobutyric acid (GABA) receptors. This is supported by the fact that eugenol administration inhibits GABA receptor currents in trigeminal ganglion neurons and suppresses GABA α1β2γ2 receptors expressed in these neurons [155,159]. 

## 5. Potential Food Application of CEO

Most EOs, including CEO, are safe for consumption, and have been generally recognized as safe (GRAS) by the Food and Drug Administration (FDA) [160]. Cloves (immature flower buds) and their preparations have also been recognized as safe by the European Food Safety Authority (EFSA) [161]. As a result, CEO is used in a wide range of food categories as a spice or for flavoring purposes (Table 3). It can infuse intense flavor and aroma into dishes, making it a popular choice for those seeking to elevate their culinary experiences with natural and aromatic ingredients. In addition, most of the individual components of clove bud oil are currently authorized as food flavorings without limitations, and have already been evaluated for consumer safety when used as feed additives in animal production [161,162]. Generally, in the food industry, cloves are often used ground, in the form of extracted essential oils, or oleoresin in small quantities due to their intense flavor. The advantages of using ground cloves are that they retain a considerable degree of their original stability during storage and tolerate high-temperature processing better than many of the extracted processed products. In this sense, clove oil is one of the most important essential oils used to flavor all kinds of food products, such as sausages, bakery products, and confectionery, among many others [163]. However, the use of CEO in food materials is complicated due to their low solvency in water. Therefore, encapsulation methods have recently been presented as effective means to extend their dispersibility in fluid media, and this technique could suggest an original methodology for applications in food and active packaging [164]. As significant and potential antiseptic in the food production, CEO, and its vitally compelling synthesis of eugenol, shows useful benefits on antibacterial and antifungal effect, safety, and aromaticity. Studies found that both CEO and eugenol express altogether inhibitory consequences for various sorts of food source microorganisms, and the mechanisms are related with diminishing the adhesion and migration, and inhibiting the synthesis, of biofilm and different harmfulness variables of these microorganisms. CEO and eugenol are, for the most part, viewed in in vivo studies, although they might express specific cytotoxicity on fibroblasts and different cells in vitro. Studies that concentrate on the quality and added substance standard of CEO ought to be reinforced to advance the antimicrobial and antioxidant impacts in the demanding food industry [62,165].

### 5.1. Food Packaging

In recent years, CEO, with its antimicrobial and antioxidant effects, can be used with polymers like proteins, lipids, and polysaccharides to form novel packaging materials to inhibit foodborne pathogens, prolong the storage time and/or improve the film’s properties [62]. Polylactic acid (PLA) and CEO have been used mainly for their antimicrobial properties in combination with graphene oxide nanosheets to also improve the tensile strength [166], with poly(ε-caprolactone) and zinc oxide (ZnO) for the storage of scrambled eggs [167], with polybutylene adipate-co-terephthalate (PBAT) for UV blocking [125], and with mesoporous silica nanoparticles for immobilization as active material [168]. Gelatin can also be used with CEO and with ZnO nanorods in shrimp packaging for higher antimicrobial effect and flexibility, but also lower UV and oxygen permeability [169], with ZnO for 3D food printing [170], and with an agar membrane and nanocellulose for higher antioxidant capacity, transparency, and UV protection [171]. Overall, food packaging has several advantages in that it provides a physical barrier that protects food from mechanical damage, contamination, light, moisture and oxygen, which helps to preserve the quality and freshness of the products. The incorporation of CEO into biodegradable films or coatings has a significant impact on their physical properties. Studies have shown that CEO enhances the water resistance, opacity, and UV blocking efficiency of these materials. Additionally, the addition of CEO can lead to improvements in tensile strength, percentage of elongation, and Young’s modulus of the films, enhancing their mechanical properties. Furthermore, other researchers used CEO with modified linear low-density polyethylene (LLDPE) by chromic acid (CA) for chicken meat packaging [126], with edible citrus pectin coating for better antimicrobial, antioxidant thermal and elastic capacity [172], with edible millet starch film for higher antimicrobial inhibition [173], and with pullulan/chitosan/ZnO films in chicken meat for oxygen, humidity, UV, and antioxidant protection [174]. Also, edible mechanically deboned chicken meat protein coatings enhanced with CEO was used in beef sucuks for its antioxidant and antimicrobial properties. MDCM-P based coating application with CEO decreased the weight loss, retarded the color deterioration, inhibited the lipid oxidation, delayed the growth of microorganisms, and improved the storage quality of heat treated sucuks in refrigerated storage [175,176]. 

**Table 3 antioxidants-13-00488-t003:** Recent studies in food application of the CEO.

Food Category	Food	Application Form/Role	Dose	Main Findings	Reference
Food packaging	PLA/graphene oxide nanosheets/CEO composite films *	Fortification	15–30%	Improved flexibility	[166]
	Gelatin/ZnO nanorods/CEO films for shrimp packaging *	Fortification	25–50%	Greater transparency and inhibition effect	[169]
	LLDPE/CA/CEO *	Fortification	-	Inhibitory effect against *L. monocytogenes* and *S. Typhimurium* for 3 weeks	[126]
	PLA-PBAT/CEO film *	Fortification	1–10%	Improved UV blocking and antimicrobial properties	[125]
	Gelatin/ZnO/CEO nanocomposite	Fortification	25%	3D food printing	[170]
	Citrus pectin film *,+	Coating	0.5–1.5%	Enhanced antimicrobial, antioxidant, mechanical, and barrier characteristics of edible pectin film	[177]
	Gelatin/agar-based film CEO stabilized with nanocellulose +	Fortification	-	Improved transparency, UV blocking, and antioxidant properties	[171]
	PLA/poly(ε-caprolactone)/ZnO/CEO films for scrambled egg packaging *	Fortification	25%	Prolonged the storage duration for up to 3 weeks, better structural and mechanical characteristics	[167]
	PLA/CEO biocomposite food packaging film *	Fortification	3%	Improved antimicrobial activity	[168]
	Mechanically deboned chicken meat protein film for beef sucuks *,+	Fortification	1%	Improved antimicrobial and antioxidant activities	[175,176]
	Millet Starch/CEO edible Film *	Fortification	1–3%	Inhibitory effect against *T. fungi*, *P. aeruginosa*, *B. cereus*, *Enterobacter* sp., *S. aureus*, and *E. coli*	[173]
	Chitosan/pullulan based films with CEO/chitosan-ZnO hybrid nanoparticles for active food packaging *,+	Fortification	1.5%	Improved UV, water, and oxygen blocking, better antimicrobial and antioxidant activity, and prolonged storage duration for up to 5 days	[174]
Meat, seafood, and poultry	Burger-like meat products *,+	Fortification	1 μL/g	Similar effects to nitrite, but better against *S. aureus*	[16]
	Pork *	Storage	0.25–8.0 mg/mL	Antibacterial effect against *S. aureus* for 1 week without texture changings	[178]
	Refrigerated bluefin tuna (*Thunnus thynnus*) fillets *,+	Coating	0.5%	Prolonged the storge duration for up to 2 weeks, decreased lipid oxidation and Pseudomonads growth	[179]
	Bream (*Megalobrama ambycephala*) +	Coating	1–1.5%	Prolonged the storge duration for up to 2 weeks, decreased lipid oxidation in cold storage	[180]
	Tambaqui (*Colossoma macropomum*) fillets *,+	Coating	0.08%	Prolonged the storge duration for up to 4 months at −18 °C	[181]
	Chinese bacon +	Fortification	1 mg/mL	Decreased lipid oxidation and increased hydroxyl and superoxide anion radical scavenging activities	[182]
	Young bulls +	Feed	450 mg/kg	Color and pH changes, decreased lipid oxidation, and the 1-week aged meat was more accepted	[183]
	Cattle +	Feed	0.5–2 g/animal/day	Improved meat quality with higher muscle to body ratio	[184]
	Chicken breast meat *	Storage	0.035–4.5 g/L	Chicken breast meat *	[185]
	Broilers +	Feed	0.05%	Improved growth performance, meat quality, and relieved gut’s integrity coccidial damage	[186]
	Pork patties *	Coating	-	Prolonged the storge duration for up to 6 days	[187]
Dairy products	Soft cheese *	Storage	0.01–0.02%	Antimicrobial properties during storage at 4 °C for 30 days, with organoleptic enhancement and storage duration up to 3 weeks	[188]
	Edam cheese *	Storage	500 μL/L	Antifungal effects against *Streptococcus* spp.	[189]
	Fresh Double Cream Cheese *,+	Storage	0.37%	Oxidation stability, antimicrobial properties against *S. aureus*, *E. coli*, and *Salmonella enteritidis*, and prolonged the storage duration for up to 13 days	[190]
Bakery products	Finger citron crisp and green bean cakes *	Storage	0.21–1.67 μL/mL	Prolonged the storge duration for 3–4 days and almost 10 days in vacuum package	[133]
	Cake *,+	Storage	0.06–0.08%	Oxidation stability and inhibitory effect for 28 days at 22 °C	[191]
	Bread *	Storage	125–500 μL/L	Alternative substances to decrease the sporulation, growth, and mycotoxins like aflatoxin and ochratoxin produced by aspergilli at 22 °C for 2 weeks in the dark	[192]
	Refrigerated steamed buns *	Coating	0–1.2%	Prolonged the storage duration for up to 10 days, yet with losses in the re-steaming cycle of volatile compounds	[193]
Fruits and vegetables	Fresh-cut lettuce +	Storage	0.05%	Anti-browning effects against browning-relevant enzymes like polyphenol oxidase, peroxidase, and phenylalanine ammonia lyase	[194]
	Apple tubes *,+	Coating	0–0.75%	Extraordinary antioxidant and antimicrobial coating at 1 °C, delaying the color and firmness spoilage	[182]
	Cut apples	Storage	0.25–1%	Prolonged the storage duration for 6 days, delaying the firmness and weight loss	[195]
	Cucumber and lettuce slices *	-	0.25–4.0 mg/mL	L. monocytogenes biofilm effectively removed	[196]
	Pak choi *	Storage	0.02%	Prolonged the storge duration to 17 days	[197]
	Persimmon *	Storage	1.56%	Inhibition of mold growth for 4 weeks	[198]
	Mango *,+	Storage	106 µL	Prolonged the storge duration to 3 weeks	[199]
	Pomegranate arils *	Coating	0.15–0.3%	Prolonged the storge duration for 54 days	[200]
	Pomegranate *	Coating	0.02–0.08%	Alternative biopesticide against *A. niger*	[201]
	Citrus Fruit *	Storage	0.05–0.8%	Inhibitory effect against Penicillium italicum	[202]
	Papaya (*Carica papaya* L.) *	-	3 mL/L	Alternative biopesticide against anthracnose disease	[203]

Bioactivity: * antimicrobial; + antioxidant.

### 5.2. Meat, Seafood, and Poultry

The use of CEO to goods from animals diminishes bothersome problems, including the decay of smell, taste, texture, color, and other organoleptic properties [62]. In pork patties, CEO has been funded to have similar antioxidant and antimicrobial effects to nitrate against *L. monocytogenes*, *E. coli*, *S. Typhimurium*, and *S. aureus* without the characteristic odor [16] and prolong the storage time at 4 °C for almost one week [187]. Li et al. [178] studied the inhibitory effects of CEO against *S. aureus* in pork, and Wang et al. [204] found the strong antioxidant effects of CEO in Chinese bacon. Regarding seafood and fish products’ preservation, CEO can improve the shelf life, texture, color, and also provide its antioxidant and antimicrobial properties in bluefin tuna (*Thunnus thynnus*), tambaqui (*Colossoma macropomum*), and bream (*Megalobrama ambycephala*) fillets [179,180,181]. CEO can be also added in livestock feed like bulls, cattle, and broilers to improve the meat quality, growth, taste, and color [183,184,186]. Last but not least, in chicken breast meat, CEO showed its anti-microbiological properties against *S. aureus*, *E. coli*, and *P. fluorescens* [185].

### 5.3. Dairy Products

Dairy product consumption has influenced different episodes of foodborne infections [62]. In fresh soft cheese model with *S. aureus* and *P. aeruginosa*, CEO showed critical antimicrobial activity due to its hydrophobicity during the capacity time of one month at 4 °C, along with an improvement in organoleptic properties [188]. Hlebová et al. [189] found the antifungal activity of CEO in vapor phase against *Penicillium commune* in Edam cheese, and [190] used CEO due to its antimicrobial properties in fresh double cream cheese to increase its antioxidant stability and shelf life for 13 days.

### 5.4. Bakery Products

The industry of baked goods is focusing on increasing the nutrition value, shelf life, and safety, but most important to avoidance the mold growth, by usually apply modified storage atmosphere, radiation, coating, and aseptic packaging [133,191,193]. There are also preservative acids like benzoic, propionic, and sorbic acids being used, even though some of them are now restricted considering the warning regarding their effects on human health [62]. CEO, and mainly eugenol, is useful against foodborne pathogenic microorganisms such as *Aspergillus* spp., *Penicillium* spp., *S. aureus*, and *E. coli* through antifungal and antibacterial mechanisms, and simultaneously does not have an impact on the sensory properties like flavor, taste, appearance, texture, or acceptability [192,205]. Ju et al. [133] found that the inhibitory effects of CEO on mold growth in green bean and finger citron crisp cakes can extend the shelf life for up to 3–4 days and, in a vacuum package, up to 9–10 days. Also, cakes containing CEO at room temperature for 28 days have shown high oxidation stability [191]. Císarová et al. [192] focused on the antifungal, anti-sporulation, and anti-toxicogenic properties of CEO using the vapor technique in bread after 2 weeks at room temperature storage, testing for *A. flavus*, *Aspergillus parasiticus*, *Aspergillus ochraceus*, and *Aspergillus westerdijkiae*. The storage time and quality also increased for refrigerated steamed buns at 10 °C with an edible coat with CEO [193], and for bread using sachets with CEO at 25 °C [205].

### 5.5. Fruits and Vegetables

Post-harvest vegetable disintegration during transport and capacity prompts critical financial misfortunes along the store network, but the antimicrobial properties of CEO can stop fungal spoilage as an alternative option to synthetic fungicides and promote a healthy and high-value diet with no changes to their organoleptic characteristics. CEO has been found to have strong inhibition effects on a variety of food-source bacteria, and its mechanisms are linked to lowering migration and adhesion, as well as blocking the creation of biofilm and various virulence factors. In the context of fruits and vegetables, CEO has been shown to delay the growth of molds and yeasts, lower weight loss and maintain firmness, and reduce microbiological changes, especially in mold and yeast growth [62]. Wang et al. [182] studied the antioxidant and antimicrobial outcome of adding CEO to chitosan as active film, and Boro et al. [195] synthesized nanocomposite films of polylactic acid (PLA) enhanced with CEO for packages of cut apples, showing better acidity, weight loss, and organoleptic properties after 6 days. Chen et al. [194] studied the multifunctional anti-browning properties of CEO in fresh-cut lettuce through polyphenol oxidase, peroxidase, and phenylalanine ammonia lyase, and Zhang et al. [196] revealed the inhibition of CEO against *L. monocytogenes* biofilm in lettuce and cucumber slices. Also, the inhibition of CEO in altered atmosphere against *S. Typhimurium* and *L. monocytogenes* extend the shelf life of pak choi up to 2 weeks [197], extended the shelf life against *A. niger* on pomegranate to 10 days [201], and against mold growth (*Rhizopus oryzae* and *A. niger*) on dried persimmons, in combination with UV-C, to almost one month [198]. The storage duration of mangos also increased to 3 weeks in an atmosphere altered with antioxidant CEO vapors [199], and increased to almost 2 months in pomegranate with a chitosan-CEO coating [200]. Lastly, CEO can be used in citrus fruits against blue mold by *P. italicum* [202], and in papaya (*Carica papaya* L.) against *Colletotrichum gloeosporioides* [203].

## 6. Encapsulation Strategies of the CEO

EOs consist of both labile and volatile compounds that have a tendency to decompose or evaporate readily under various conditions such as processing, utilization, and storage. Factors such as high temperatures, low pressures, exposure to air and light, among others, can contribute to the decomposition or evaporation of these compounds when EOs are incorporated into foods or packaging materials [164]. The encapsulation of bioactive compounds like EOs proves to be a highly effective method not only for shielding them from degradation in unfavorable environmental conditions, but also for prolonging the shelf life of EOs. Additionally, encapsulation facilitates the development of controlled release delivery systems for these compounds [206]. Numerous methods for nanoencapsulation of bioactive compounds have undergone thorough investigation using different types of shell materials [207]. Shell materials refer to substances employed to encapsulate sensitive bioconstituents or EOs, shielding them from adverse conditions such as air exposure, light, temperature variations, humidity, and the rigors of food processing. This protective encapsulation serves to prevent volatilization, oxidation, instability, and insolubility of the EOs [208].

CEOs stands out among EOs for its capacity to manage postharvest contamination in diverse agricultural commodities, including cereals, oilseeds, fruits, and nuts. This is particularly crucial in preventing the accumulation of ochratoxin A, a significant mycotoxin produced by plant pathogens like *A. niger*. Ochratoxin A is linked to conditions such as carcinoma, nephropathy, and immunosuppressive diseases [209]. Its application as a food preservative is attributed to the multifaceted properties of CEO, encompassing antibacterial, antifungal, antioxidant, insecticidal, and antiviral attributes (Table 4). Nonetheless, the antimicrobial and antioxidant activity of CEO is notably restricted by its constituents, such as eugenol, which are highly volatile and exhibit limited solubility in water [200]. 

Enclosing EOs within micro- and nanoparticles (MPs and NPs), micro- and nano-capsules (MCs and NCs), films, or nanocomposite materials has been proposed. These methodologies enhance the stability of CEO in aqueous environments, subsequently improving its bioavailability, mitigating potential adverse effects, providing controlled release of the encapsulated substance, offering protection from the external environment, and concealing its strong odor [210]. Micro- and nano-carriers with tailored characteristics are particularly intriguing due to their larger surface-to-volume ratios, leading to heightened reactivity. Generally, these systems are constructed from lipids, polymers, or a combination of both. Moreover, distinctions exist between micro- and nanocarriers concerning their post-application behavior, their capability to traverse specific biological barriers, their efficacy in cellular entry, and potential tissue responses. These factors play a pivotal role in determining the preferable choice between the two depending on the intended application [86]. The utilization of encapsulation has been instrumental in enhancing the overall value of CEO by extending its shelf life, improving its physicochemical stability, facilitating controlled release, and proposing novel applications. The literature contains a multitude of documented encapsulation techniques employing diverse carriers, yielding intriguing results. The final products can manifest as emulsions, complexes, liposomes, micelles, or particles/capsules (Table 4). Most of the research has emphasized the stability and improved bioavailability of CEO, along with the preservation of its valuable biological properties during processing and storage. Simultaneously, other studies have directed their attention toward its effectiveness against specific strains of microorganisms [22].

Addressing these challenges, numerous studies have assessed the impact of CEO encapsulation on both antioxidant and antimicrobial activities. Sharma et al. [211] formulated an active packaging system, employing a vapor-phase antimicrobial agent embedded within chitosan capsules. This innovative approach aimed to prolong the shelf life of dry cakes, with the encapsulation of CEO achieved through an emulsion-ionic gelation crosslinking technique. Various proportions of CEO loading in chitosan were utilized, including ratios of 0.0:1, 0.25:1, 0.50:1, 0.75:1, 1:1, 1.25:1, and 1.50:1. The evaluation of antimicrobial activity in the vapor phase was conducted using active capsules. Complete prevention of *E. coli* and *S. aureus* development was achieved by employing CEO with chitosan in a ratio exceeding 1:1. Ultimately, dry cakes packed with these active capsules experienced a reduction in bacterial growth, extending during 10 days of packing. In alignment with the previous study, Fang et at. [212] aimed to document the nanoencapsulation of CEO as a natural food preservative. They explored its augmented antimicrobial properties through the utilization of a 3D nanonetwork porous starch-based material (3D-NPS). For the initial time, 3D-NPS loaded with CEO was crafted using a sacrificial template technique employing potato starch. The study delved into the influence of CaCO_3_ NPs concentration, encapsulation efficiency, temperature, and antimicrobial activity on the characteristics of CEO-loaded 3D-NPS. Following the encapsulation of CEO, the CEO-loaded 3D-NPS achieved a peak encapsulation efficiency of 86.7% and demonstrated improved stability under thermal treatment. Additionally, the 3D-NPS loaded with CEO exhibited significantly heightened antimicrobial efficacy against *B. subtilis*, *S. aureus*, and *E. coli*, and notably prolonged the duration of CEO’s antimicrobial activity. Similarly, Adjali et al. [213] introduced the encapsulation of CEO using hydroxypropyl-β-cyclodextrin (HP-β-CD), followed by the integration of the resulting inclusion complex into a flexible chitosan film. The objective is to attain a controlled release profile for the volatile CEO. The DPPH assay was employed to evaluate the antioxidant activity of both free CEO (71%) and the CEO-HP-β-CD inclusion complex. The results highlighted an improved radical scavenging ability of the EO following its encapsulation in HP-β-CD, reaching 87%. 

Nirmala et al. [214] examined the anticancer and antibacterial effects of a nanoscale emulsion system based on clove essential oil. In this study, the standard titration technique was used to prepare the microemulsions. The CEO was titrated against the surfactant aqueous phase at laboratory temperature to prepare formulations with different oil:surfactant ratios, from 1:1 to 1:9, and from 1:9 to 9:1. The results showed the efficacy of the emulsion system in inhibiting cancer cell growth and antibacterial activity against various bacterial strains. 

**Table 4 antioxidants-13-00488-t004:** An overview of encapsulation techniques for CEO.

Extraction Technique	Encapsulation Method	Key Finding	References
-	Complex coacervation	The encapsulation of CEO exhibited higher antimicrobial activity compared to the unencapsulated form.	[215]
-	Emulsion-ionic gelation crosslinking technique	The complete prevention of *E. coli* and *S. aureus* development was achieved by utilizing CEO.	[211]
Clevenger apparatus	Inclusion complex	The antioxidant activity was significantly boosted to 88% with the encapsulated CEO, surpassing the 71% observed with the free form of CEO.	[213]
SD	Emulsification	The encapsulated CEO demonstrated potent fungicidal effects against *B. cinerea*, preventing both infection and the development of disease.	[216]
-	Freeze-drying	The encapsulation of CEO resulted in a noteworthy augmentation of the total antioxidant activity, phenolic content, antifungal activity, antioxidant capacity, and oxidative stability.	[217]
-	Electrospinning	The constructed mat demonstrated effective antibacterial activity against *S. aureus* and *E. coli*, along with non-cytotoxic behavior towards human fibroblast cell lines. Additionally, it exhibited promising potential for wound healing.	[218]
Hydrodistillation using a Clevenger-type	Emulsification	Encapsulation increased the antioxidant and antifungal properties of CEO.	[219]
-	Emulsification and 3D printing	Encapsulation greatly enhanced the antimicrobial activity of CEO.	[220]
-	Sol-gel process	The findings suggested that the antifungal efficacy tripled in the encapsulated form.	[221]
-	Emulsification	The encapsulated CEO showed high antifungal activity and the quality of pomegranate arils.	[200]
Water distillation technique	The encapsulation done in two steps. First oil-in-water emulsification and second, ionic gelation method.	Encapsulation resulted in an enhancement of the antioxidant and antibacterial properties of CEO.	[164]
-	Emulsion-ionic gelation technique	The controlled release of oil led to an enhancement in the antifungal activity of the encapsulated oil.	[131]
SD using the Clevenger equipment	Emulsification and air circulation oven	Encapsulated CEO exhibits robust antimicrobial and antioxidant properties. Through the encapsulation process, the distinctive aroma of CEO was restrained.	[16]
-	Kneading and freeze-drying	The encapsulation process elevated both the total phenolic content and antioxidant activity of CEO.	[221]
-	Sonication	Encapsulation of CEO in chitosan-myristic acid enhanced the antibacterial and antioxidant activity. The applied coating exhibited increased antibacterial efficacy against *Salmonella enterica* Ser. Enteritidis.	[222]
Extracted using 95% ethanol (*v*/*v*)	Emulsification	The encapsulation of CEO/cinnamon demonstrated a higher antimicrobial activity against the common microorganisms.	[223]
-	Liposome	The efficient inhibition of *S. aureus* in tofu was achieved through the use of liposome-encapsulated clove oil.	[224]
-	Ethanol injection method	The encapsulated CEO demonstrated higher antioxidant activity.	[225]
Supercritical carbon dioxide (SC-CO_2_) extraction	Spray drying	Encapsulation improved the antioxidant activity of CEO.	[226]

## 7. Concluding Remarks and Future Perspectives

This paper reviews the phytochemistry, extraction methods, bioactivity, encapsulation techniques, and applications of CEO and its main components. The bioactive substances of CEO can be extracted using conventional techniques, such as hydrodistillation and steam distillation, or through more novel and selective techniques, including supercritical fluid extraction, ultrasound-assisted extraction, microwave-assisted extraction, among others. 

Eugenol, β-caryophyllene, α-humulene, and eugenol acetate are the main valuable compounds of CEO with antioxidant, antibacterial, anti-inflammatory, anti-hypertensive, and anticancer properties. Especially the antioxidant and antimicrobial properties of CEO encourage its use in the food industry for meat, poultry, seafood, vegetables, dairy products, and food coatings. Nevertheless, the clove origin, agro-ecological conditions, and processing and extraction methods may qualitatively and quantitatively modify the chemical composition of the CEO.

Although the consumption and use of CEO is high, there is still a lot of research to be done. Further research is needed to determine the role of key constituents in various biological processes for possible use in the treatment of various diseases. In addition, it is also relevant to know whether there is any synergism or antagonism between CEO components. Also, there is a need to study the use of clove oil in the food industry, especially as an antioxidant or antimicrobial without adverse effects on the color, taste, odor, and texture of food products. Finally, despite all the research, some encapsulation materials and potential applications have not yet been thoroughly investigated. For this reason, new studies could contribute to understanding the role of CEO in the treatment of diseases, which would allow the development of future applications in the fields of medicine, food, cosmetics, and other industries.

## Figures and Tables

**Figure 1 antioxidants-13-00488-f001:**
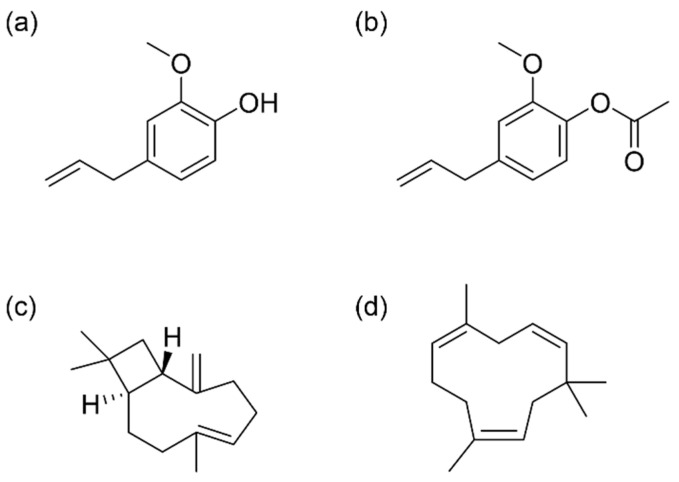
Main bioactive compounds identified in CEO. (**a**) Eugenol; (**b**) Eugenol acetate; (**c**) (E)-β-caryophyllene; (**d**) α-humulene.

**Figure 2 antioxidants-13-00488-f002:**
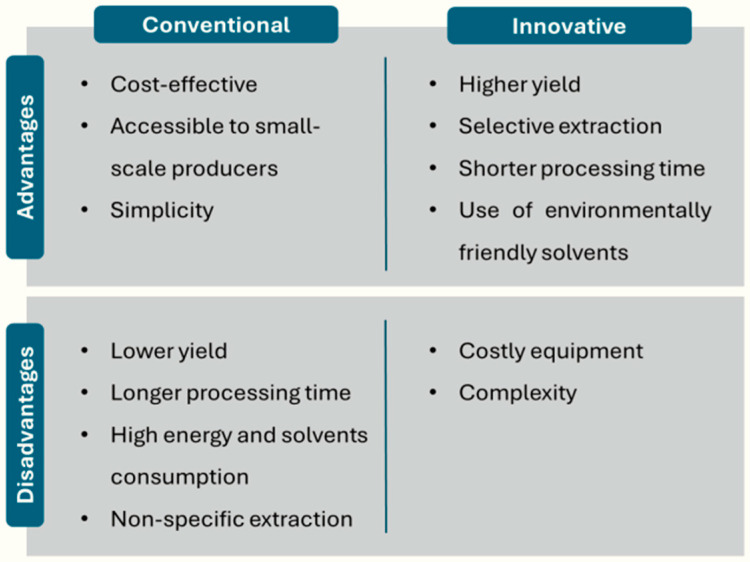
Advantages and disadvantages of conventional and innovative extraction methods.

**Figure 3 antioxidants-13-00488-f003:**
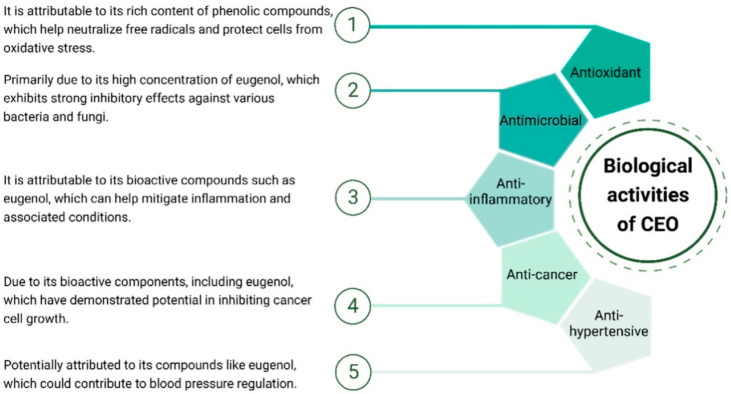
Biological activities of CEO.

**Figure 4 antioxidants-13-00488-f004:**
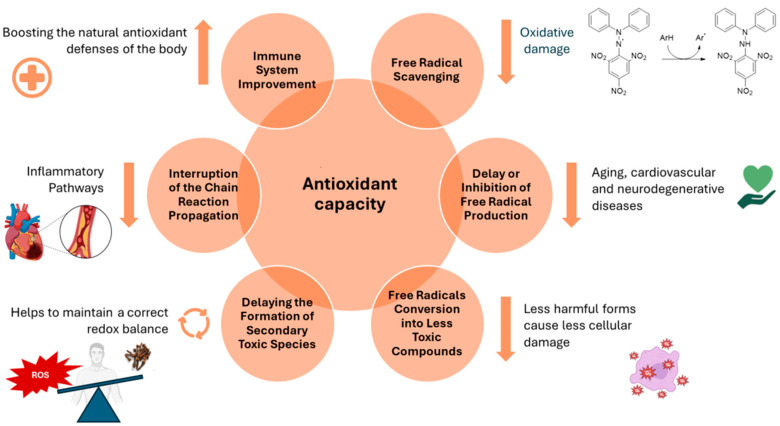
Main mechanisms of antioxidant action.

**Figure 5 antioxidants-13-00488-f005:**
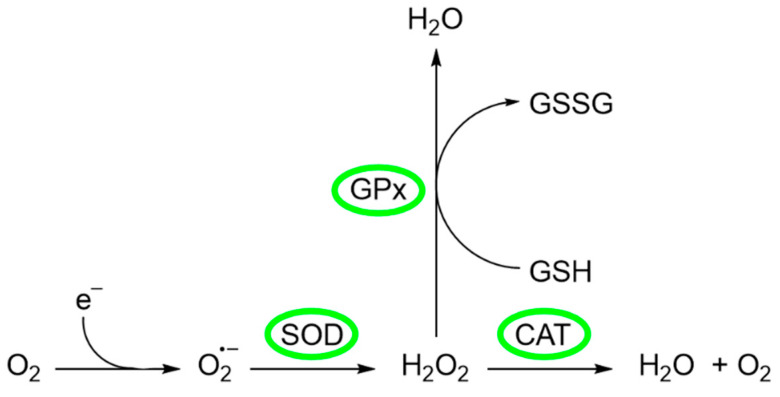
Generation and metabolism of ROS.

**Table 1 antioxidants-13-00488-t001:** The main compounds of CEO based on the most recent studies.

References		[25]	[26]	[27]	[28]
Geographic location		Shanghai, China	Marrakech, Morocco	São Paulo, Brazil	Nitra, Slovakia
Extraction technique		SFE	HD	HD	SD
	Compound	Content (%)			
1	Eugenol	55.28	82.16	97.98	82.40
2	(E)-β-caryophyllene	20.97	0.79	0.11	14.00
3	Eugenol acetate	-	16.55	1.01	0.90
4	α-humulene	7.08	-	-	1.80
5	Caryophyllene oxide	1.25	-	-	0.70
6	4-Hydroxy-2-methoxycinnamaldehyde	1.08	-	-	-
7	1-thienylcyclohexene	0.63	-	-	-
8	Isoaromadendrene epoxide	0.40	-	-	-
9	α-farnesene	0.36	-	-	-
10	Camphor	-	0.29	-	-
11	Camphene	-	0.21	-	-
12	4-methyl-1,3-Dioxolane	0.04	-	-	-
13	4-(2-propenyl)-Phenol	0.01	-	-	-
	Total	87.1	100	99.1	99.8

- Not detected.

**Table 2 antioxidants-13-00488-t002:** Conventional and innovative extraction techniques of CEO.

Extraction Technique	Condition	Sample	CEO Yield (%)	Reference
SD	Time: 8–10 h Mass: 100 g	Clove buds	10.10	[18]
HD	Time: 4–6 h Mass: 100 g	Clove buds	11.50	[18]
SE	Solvent: absolute ethanol Time: 6 h Mass: 30 g	Clove buds	41.80	[18]
SFE	Temperature: 50 °C Pressure: 10 MPa Mass: 15 g	Clove buds	19.56	[18]
SE	Solvent: n-hexane Time: 6 h Mass: 10 g	Clove leaves	1.90	[64]
SFE	Temperature: 40 °C Pressure: 220 bar Time: 80 min Mass: 18 g	Clove leaves	1.08	[64]
SFE	Time: 210 min Temperature: 44.7 °C Pressure: 24.5 MPa Mass: 30 g	Clove buds	17.90	[65]
SFE	Time: 20 min Temperature: 40 °C Pressure: 15 MPa Mass: 12 g	Clove buds	21.30	[66]
SFE-cold pressing	Time: 20 min Temperature: 40 °C Pressure: 15 MPa Torque: 40 N·m Mass: 12 g	Clove buds	22.20	[66]
UAE-CO_2_	Time: 30 min Temperature: 40 °C Pressure: 15 MPa Mass: 5 g	Clove buds	22.04	[67]
UAE-CO_2_	Time: 2 h Temperature: 44 °C Pressure: 28.5 MPa Mass: 20 g	Clove buds	23.19	[68]
UAE-HD	Power: 750 W Frequency: 20 kHz Amplitude: 40% Time: 50 min Mass: 15 g	Clove buds	15.23	[60]
MAE	Power: 600 W Time: 30 min Volume of water: 200 mL Mass: 30 g	Clove buds	13.11	[21]
MAE-HD	Power: 1000 W Time: 80 min Ratio solid/water: 1/10	Clove buds	13.94	[69]
MAE-SD	Power: 1000 W Time: 80 min Ratio solid/water: 1/10	Clove buds	12.71	[69]
MAE	Time: 90 min Ratio solid/water: 1:40 Mass: 40 g	Clove buds	16.00	[63]
OAHD	Voltage gradient: 12.5 V/cm Time: 97.967 min Mass: 40 g	Clove buds	13.18	[20]
HD	Solvent: Distilled water Time: 97.53 min Mass: 40 g	Clove buds	8.23	[20]

## Data Availability

The data presented in this study are available in the article.

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
