# Peer review of "Clove Essential Oil: Chemical Profile, Biological Activities, Encapsulation Strategies, and Food Applications"

_antioxidants, 2024, doi:10.3390/antiox13040488_

Round 1
Reviewer 1 Report (Previous Reviewer 1)
The authors improved the manuscript and it can be accepted after minor revisions.
The manuscript can be accepted after minor revisions. Please see my comments to improve the article.
Section 5, please introduce the CEO as an aroma/flavor in the food matrix, the security aspect of the CEO should be introduced.
Section 5.1, please introduce more citations regarding the physical properties of the packaging. Also, the migration from packaging to food should be addressed.
Section 5.5, please cite the articles introducing how CEO on fruit and vegetable quality.
Section 6 lacks a deep review of the encapsulation strategy of the CEO. Some important encapsulation methods should be supplied, such as emulsions.
Author Response
Reviewer 1
1- The authors improved the manuscript and it can be accepted after minor revisions. The manuscript can be accepted after minor revisions. Please see my comments to improve the article.
Answer: We would like to thank you for the time and effort taken to review our manuscript, we have carefully responded to each comment. Your requested corrections carried out in the track changes.
2- Section 5, please introduce the CEO as an aroma/flavor in the food matrix, the security aspect of the CEO should be introduced.
Answer: It has been added in the section 5 based on your comment.
3- Section 5.1, please introduce more citations regarding the physical properties of the packaging. Also, the migration from packaging to food should be addressed.
Answer: It has been revised.
4- Section 5.5, please cite the articles introducing how CEO on fruit and vegetable quality.
Answer: The mechanisms of effect of CEO on fruit and vegtables quality have been added in this section (Section 5.5).
5- Section 6 lacks a deep review of the encapsulation strategy of the CEO. Some important encapsulation methods should be supplied, such as emulsions.
Answer: More studies have been added to this section to improve encapsulation strategy of the CEO. Also, more details about encapsulation methods has been mentioned. We try to cover all recent studies about encapsulation strategy of CEO in the manuscrpit.
Reviewer 2 Report (New Reviewer)
The topic of the paper is extremely extensively covered and talks about the biological potential and application of CEO. The paper can be written in more detail about essential oil isolation techniques, which are the advantages of modern extraction techniques, not only in relation to the yield of essential oil and differences in chemical composition, but also to environmental protection due to the smaller amount of use of organic solvents, saving time , electricity, water reduction...
Comments and suggestions for Authors
Dear authors, I reviewed in detail the paper entitled "Clove essential oil: Biological activities, encapsulation strategies and food applications".
These are my comments and suggestions:
The title should be corrected: insert the chemical profile as well
Line 33-34: antioxidant activity?, activity missing. It is better to write biological activity in keywords.
Lines 103-104, line 146 and 192-193: not beta-caryophyllene, alpha-humulene, but b- caryophyllene and a- humulene
Lines 252-256: Move this sentences to the introduction.
Write down in the article the advantages and disadvantages of classic and advanced techniques for isolating essential oils, also including in terms of environmental acceptability.
Latin names should be in italic
Some parts of manuscript are in red (font color)? I don’t understand why. Correct this.
Line 808: Instead of anticancer activity, it is better to write cytotoxic activity if we are talking about in vitro studies.
Line 992: not wight, but weight
Figures and tables are clear and well presented.
Write references according to the instructions for authors.
The conclusion is written in accordance with the topic covered in the review artice. The manuscript is interesting, but very extensive, which is why I think it should be reorganized some parts of manuscript and better structured so that there is no repetition of data (especially thinking of section 2 “Main bioactive substances of the CEO” where you already mention biological activities that show the components present in CEO, and then below you have sections related to biological activity), English is understandable. I suggest acceptance after accepting all comments. I suggest acceptance after minor revision.
Kind regards

Author Response
1- The topic of the paper is extremely extensively covered and talks about the biological potential and application of CEO. The paper can be written in more detail about essential oil isolation techniques, which are the advantages of modern extraction techniques, not only in relation to the yield of essential oil and differences in chemical composition, but also to environmental protection due to the smaller amount of use of organic solvents, saving time , electricity, water reduction...
Answer: We would like to thank you for the time and effort taken to review our manuscript, we have carefully responded to each comment. Your requested corrections carried out in the track changes.
2- Dear authors, I reviewed in detail the paper entitled "Clove essential oil: Biological activities, encapsulation strategies and food applications".
These are my comments and suggestions:
The title should be corrected: insert the chemical profile as well
Answer: The title has been revised according to your suggestion.
3- Line 33-34: antioxidant activity?, activity missing. It is better to write biological activity in keywords.
Answer: It has been modified.
4- Lines 103-104, line 146 and 192-193: not beta-caryophyllene, alpha-humulene, but b- caryophyllene and a- humulene
Answer: They have been corrected.
5- Lines 252-256: Move this sentences to the introduction.
Answer: It has been moved to the introduction.
6- Write down in the article the advantages and disadvantages of classic and advanced techniques for isolating essential oils, also including in terms of environmental acceptability.
Answer: A new figure (Figure 2) has been added to the manuscript to summerize the advantages and disadvantages of classic and advanced techniques for isolating essential oils.
7- Latin names should be in italic
Answer: It has been revised.
8- Some parts of manuscript are in red (font color)? I don’t understand why. Correct this.
Answer: Sorry, they have been corrected.
9- Line 808: Instead of anticancer activity, it is better to write cytotoxic activity if we are talking about in vitro studies.
Answer: It has been revised according to your great suggestion.
10- Line 992: not wight, but weight
Answer: It has been corrected.
11- Figures and tables are clear and well presented.
Answer: Thank you for your feedback.
12- Write references according to the instructions for authors.
Answer: All the references have been double checked according to the instructions for authors.
13- The conclusion is written in accordance with the topic covered in the review artice.
Answer: Thank you for your feedback.
14- The manuscript is interesting, but very extensive, which is why I think it should be reorganized some parts of manuscript and better structured so that there is no repetition of data (especially thinking of section 2 “Main bioactive substances of the CEO” where you already mention biological activities that show the components present in CEO, and then below you have sections related to biological activity), English is understandable. I suggest acceptance after accepting all comments. I suggest acceptance after minor revision.
Answer: Sections 2 and 4 may seem repetitive, but they indicate different information that we would like to clarify to resolve any possible misunderstanding.
Section 2: "Main bioactive substances of CEO" describes on the one hand, the main compounds found in CEO and, on the other hand, the bioactivity related to these compounds. For the writing of this section, information on the biological activity of eugenol, acetyleugenol, alpha-humulene and beta-caryophyllene has been searched. However, the information that appears only refers to the bioactivity of these compounds, not to the CEO. On the other hand, section 4: "biological activities of CEO" describes the bioactivity attributed to CEO. This bioactivity is the result of the synergy of the different compounds that compose CEO. Although both sections might appear to be repetitive, they describe different information. There are other papers that also make this distinction (doi.org/10.3390/molecules27134251; https://doi.org/10.1016/j.sajb.2021.09.026). Nevertheless, a different title has been proposed for section 2 in order to avoid confusion between these sections.
This manuscript is a resubmission of an earlier submission. The following is a list of the peer review reports and author responses from that submission.
Round 1
Reviewer 1 Report
The article entitled 'Clove essential oil: Biological activities, encapsulation strategies and food applications' by Liñán-Atero et al. reviewed the biological characteristics and encapsulation technology of clove essential oil and its potential in the food field. In my opinion, the manuscript can be accepted after minor revision, following are some comments for the authors to improve the manuscript.
Table 2, I recommend showing the full name of Extraction techniques in the table captions.
Line 558, what are poison food techniques?
Table 3, the authors should double-check the unit of the doses, the percentage should be notified v/v or w/w or other. The main findings should be more specific, like line 174-176, the authors should provide the temperature and relative humidity.
Table 4, I recommend the authors supply the characteristic method of the references, which will help a lot for other researchers, and may a supply for the key findings, such as improved physiological of chemical properties.
Line 834, in this section, the authors should also discuss the controlled release effects by encapsulation strategies.
Line 911, in this section, I recommend the authors discuss the potential synergistic effect of clove essential oil with commercial sterilant or other essential oils.